# Two-Stream Network for Sign Language Recognition and Translation

**Yutong Chen**[1]*  **Ronglai Zuo**[2]*  **Fangyun Wei**[1]*†  **Yu Wu**[1]  **Shujie Liu**[1]  **Brian Mak**[2]

[1]Microsoft Research Asia  [2]The Hong Kong University of Science and Technology

chenytjudy@gmail.com  {rzuo,mak}@cse.ust.hk

{fawe,yuwu1,shujliu}@microsoft.com

## Abstract

Sign languages are visual languages using manual articulations and non-manual elements to convey information. For sign language recognition and translation, the majority of existing approaches directly encode RGB videos into hidden representations. RGB videos, however, are raw signals with substantial visual redundancy, leading the encoder to overlook the key information for sign language understanding. To mitigate this problem and better incorporate domain knowledge, such as handshape and body movement, we introduce a dual visual encoder containing two separate streams to model both the raw videos and the keypoint sequences generated by an off-the-shelf keypoint estimator. To make the two streams interact with each other, we explore a variety of techniques, including bidirectional lateral connection, sign pyramid network with auxiliary supervision, and frame-level self-distillation. The resulting model is called TwoStream-SLR, which is competent for sign language recognition (SLR). TwoStream-SLR is extended to a sign language translation (SLT) model, TwoStream-SLT, by simply attaching an extra translation network. Experimentally, our TwoStream-SLR and TwoStream-SLT achieve state-of-the-art performance on SLR and SLT tasks across a series of datasets including Phoenix-2014, Phoenix-2014T, and CSL-Daily.

## 1 Introduction

Sign languages, which are the primary means of communication among the deaf and hard-of-hearing people, have been studied for a long time [46, 45, 4]. In linguistic terms, sign languages are as rich and complex as any spoken language [2] and their word-order typology may differ from the spoken languages. Moreover, there are limited parallel data for sign-to-text, and the gap between the visual modality and the language modality poses another challenge to developing a well-performing sign language translation (SLT) system [6]. To mitigate the problem, SLT often requires an intermediate representation between an input visual signal and the output text, which is a sequence of glosses[1], to achieve satisfactory translation results. The process of generating gloss sequences from given sign videos is termed as sign language recognition (SLR) [13], which does not have the word ordering problem in SLT. Figure 1a illustrates the relationship between SLR [27, 56, 37, 11, 41, 14] and SLT [8, 6, 31, 55, 10, 56]. In this paper, we work on both tasks.

The key to tackle SLR and SLT is to build a visual encoder to embed visual signals into hidden representations. Inspired by the promising progress of action recognition [22, 48, 9, 51, 35, 3], a majority of works [36, 20, 41, 11, 37, 5] explore to directly model RGB videos to understand sign languages. However, a major problem with this kind of models is their robustness, and the models

---

*Equal contribution.

†Corresponding author.

[1]Glosses are the word-for-word transcription of sign language where each gloss is a unique label for a sign.

36th Conference on Neural Information Processing Systems (NeurIPS 2022).

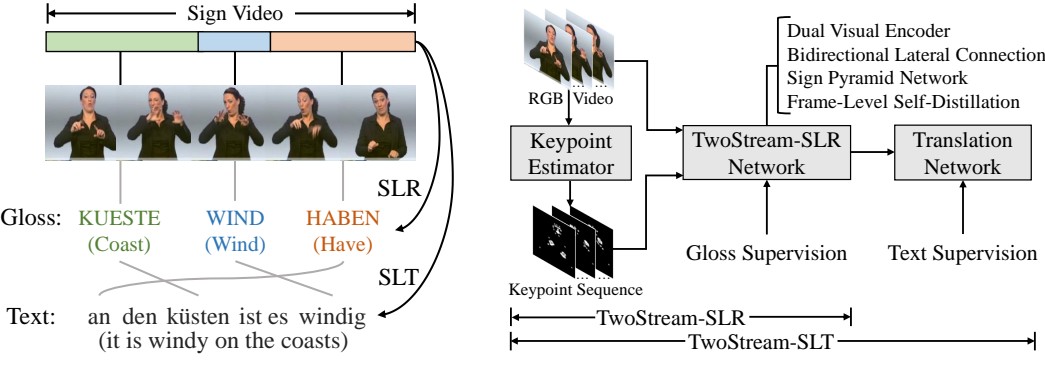

(a) Illustration of SLR and SLT.          (b) Overview of TwoStream-SLR and TwoStream-SLT.

Figure 1: (a) We select a sign video from the Phoenix-2014T [6] dataset and visualize its gloss sequence and the corresponding text. Sign language recognition (SLR) seeks to train a model to recognize a gloss sequence from a sign video with weak sentence-level gloss annotations (*i.e.*, gloss temporal boundaries are unknown). In contrast, sign language translation (SLT) aims to directly generate spoken languages (texts) with or without intermediate representations (glosses). (b) TwoStream-SLT is built upon TwoStream-SLR to enable SLT. We use HRNet [50] trained on COCO-WholeBody [23] to extract keypoints of the face, hands, and upper body. Keypoints are represented by heatmaps.

suffer from dramatic performance degradation when the background or signer is mismatched between training and testing. To alleviate the problem, we consider injecting proper domain knowledge (characteristic of sign languages) into learning. Sign languages use two types of visual signals to convey information [26]: manual elements that include handshape, palm orientation, etc., and non-manual markers such as facial expressions and movement of the body, head, mouth, eyes, and eyebrows. In this paper, we advocate involving keypoints of the face, hands, and upper body in the learning, and introduce a novel two-stream network named TwoStream-SLR to model both RGB videos and keypoint sequences for sign language recognition (SLR). The proposed TwoStream-SLR can be extended to handle sign language translation (SLT) by attaching an extra translation network. The resulting model is termed as TwoStream-SLT. An overview of TwoStream-SLR and TwoStream-SLT is shown in Figure 1b. Our contributions mainly lie in the proposed TwoStream-SLR; we summarize our design principles and key components of TwoStream-SLR as follows:

1. **Dual Visual Encoder.** We use two separate S3D [51] backbones with lightweight head networks to encode RGB videos and keypoint sequences which are represented by a set of heatmaps. Since most sign language datasets do not provide keypoint annotations, we use an off-the-shelf keypoint estimator, HRNet [50] which is trained on COCO-WholeBody [23], to generate pseudo keypoints of face, hands, and upper body for each frame. Both streams are supervised by the well-known connectionist temporal classification (CTC) loss [19]. In contrast to existing methods which either utilize keypoints to crop concerned areas on feature maps [56]/original videos [38], or provide supervision for multi-task learning [56, 1], we directly apply a convolutional neural network to model keypoint sequences to avoid ad-hoc design [25, 47]. Our dual encoder architecture is different from two-stream networks for action recognition which encode either image (video)/optical flow [44, 9], or two videos of different frame rates [17]. We fuse different streams at late stage by averaging their predicted frame-wise gloss probabilities before feeding them to a CTC decoder to produce the final gloss sequences.

2. **Information Interaction via Bidirectional Lateral Connection.** Constrained by the photographic apparatus, motion blur heavily exists in sign videos [27, 6], resulting in inaccurate keypoint estimation. This motivates us to use the information extracted by the video stream to alleviate the negative impacts caused by inferior keypoints when modeling keypoint sequences. On the other hand, heavy redundancy and irrelevant information (*e.g.*, background and appearance of signer) in sign videos may lead the model to overlook the salient information. Thus, we propose to integrate the features extracted by the keypoint stream into the video stream as supplementary information. Thanks to the dual encoder design, intermediate representations

of video and keypoint sequence streams could be easily integrated via a bidirectional lateral connection module [15] for information exchange.

3. **Alleviate Data Scarcity via Sign Pyramid Network and Auxiliary Loss.** Both SLR and SLT greatly suffer from data scarcity. For example, there are only around $7K$ parallel training samples in the widely used Phoenix-2014T [6] dataset. In contrast, training an effective neural machine translation model usually requires a corpus of $1M$ parallel data [43]. Besides, similar to actions with different visual tempos in the action recognition field [52], glosses also have different temporal spans [42]. Thus, we present a sign pyramid network (SPN), which is inspired by the feature pyramid network [33] in object detection and the temporal pyramid network [52] in action recognition, on top of the dual encoder to better capture glosses of various temporal spans in the low-data regime. The fused features yielded by SPN are further supervised by auxiliary CTC losses, which enable the shallow layers to learn meaningful features.

4. **Frame-Level Self-Distillation.** We treat the averaged ensemble predictions as pseudo-targets and propose an additional self-distillation loss calculated by the KL-divergence between each stream's predictions and pseudo-targets at the frame level. The self-distillation feeds rich knowledge in the late ensemble predictions back into each stream's learning. Moreover, compared with the CTC loss which only provides sentence-level supervision without temporal boundary information, the self-distillation loss is computed per frame to facilitate training with extra pseudo frame-level supervision.

To enable sign language translation, we append a translation network to the TwoStream-SLR, yielding our translation model named TwoStream-SLT. The proposed TwoStream-SLR and TwoStream-SLT achieve state-of-the-art performance on both SLR and SLT across a series of benchmarks including Phoenix-2014 [27], Phoenix-2014T [6], and the recently published CSL-Daily [55].

## 2  Related Work

**Sign Language Recognition and Translation.** Sign language recognition (SLR) aims to transcribe an input sign video into a gloss sequence. An SLR model usually consists of two components, a video encoder that extracts frame-level features from an input video, and a decoder (or head network) that outputs gloss sequences from the extracted features. The video encoder is usually based on CNNs including 3D-CNNs [10, 40, 31], 2D-CNNs [5, 37], and 2D+1D CNNs [56, 14], and both single-stream [10, 40, 31] and multi-stream architectures [56, 14, 38] have been adopted. In this work, we utilize two separate S3D [51] backbones to model both RGB videos and keypoint sequences. In the design of gloss decoder, all recent works use either HMM [28–30] or CTC [11, 56, 36] following their success in automatic speech recognition. We adopt CTC due to its simplicity. Since CTC loss just provides weak sentence-level supervision, [14, 54] propose to iteratively generate fine-grained pseudo labels from CTC outputs to introduce stronger frame-level supervision, while [36] achieves frame-level knowledge distillation between the entire model and the visual encoder. In this work, as a byproduct of the two-stream architecture, our frame-level distillation imparts the final ensemble knowledge into each individual stream and enhances the interaction and consistency between the two streams. Sign language translation (SLT) directly predicts texts given sign videos. Most existing approaches formulate this task as a neural machine translation (NMT) problem by employing a visual encoder as tokenizer to extract visual features and forwarding them to a translation network for spoken text generation [10, 8, 31, 55, 53, 6]. We use mBART [34] as our translation network due to its excellent SLT performance [10]. To achieve satisfactory results, gloss supervision is commonly utilized in SLT by pretraining the visual encoder on SLR [8, 56, 55] and jointly training SLR and SLT [56, 8].

**Introduce Keypoints into SLR and SLT.** How to leverage keypoints to boost the performance of SLR and SLT is still an open problem. [56] and [38] utilize estimated keypoint coordinates to crop feature maps and RGB videos to process each cue (hands, face, and body) independently. Some other works [25, 7, 47] model keypoints from coordinates. For example, the SLT system in [25] feeds keypoint coordinates into an MLP followed by recurrent neural networks. [47] represents keypoints as skeleton graphs, which are further modeled by graph convolutional networks (GCNs). However, all of the existing works treat keypoints as a set of coordinates, which is so sensitive to noise that a small perturbation may lead to wrong predictions [15]. Besides, extra efforts need to be paid to devise dedicated modules, *e.g.*, GCN. In this work, we represent keypoints as heatmaps, which are robust to noise and can simply share the same architecture with the video stream.

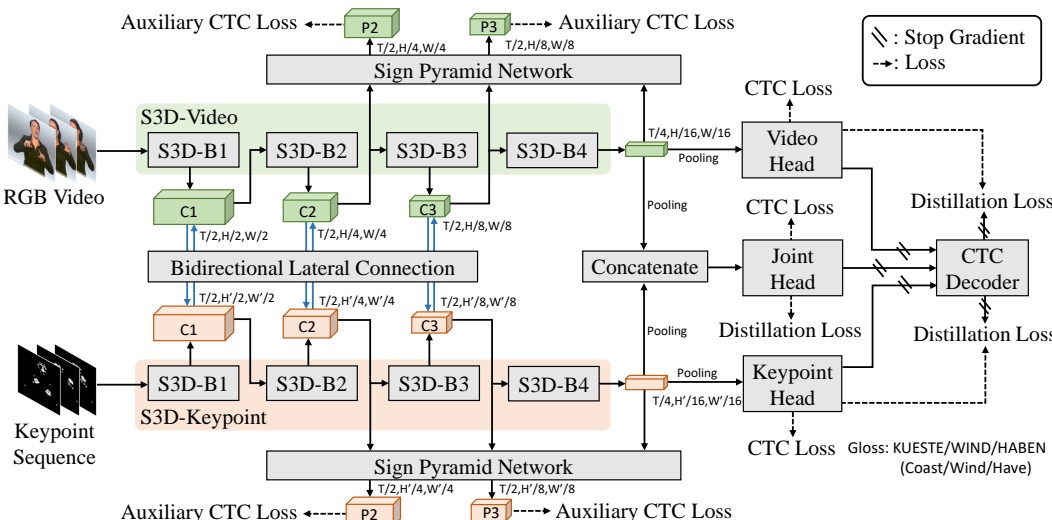

Figure 2: Overview of TwoStream-SLR, which is composed of: 1) a video encoder; 2) a keypoint sequence encoder; 3) a joint head; 4) a bidirectional lateral connection module; 5) two sign pyramid networks. The whole network is jointly supervised by the CTC losses and the frame-level self-distillation losses. Keypoint sequences are represented by heatmaps.

**Multi-Stream Networks.** Multi-Stream networks are widely explored in the action recognition field [44, 16, 9, 57, 17]. For example, I3D [9] builds a two-stream 3D-CNN architecture and takes RGB video and optical flow as inputs. SlowFast [17] encodes videos with different frame rates. As for SLR and SLT, DNF [14] follows the idea of I3D to fuse information of RGB videos and optical flow, while STMC [56] models the multi-cue property of sign languages with a multi-stream architecture that takes cropped feature maps as inputs. In this work, our approach directly models RGB videos and keypoint heatmaps via a dual encoder. Besides, to alleviate the data scarcity issue and better capture glosses of various temporal spans, we present a sign pyramid network equipped with auxiliary supervision to drive the shallow layers to learn meaningful representations. How to model the interactions between different streams is non-trivial. I3D [9] uses a late fusion strategy by simply averaging the predictions of two streams. Another way is to fuse the intermediate features of each stream in early stage by lateral connections [17], concatenation [56], or addition [14]. In this work, we extend lateral connections in [17] to bidirectional ones to make two streams complement each other. In addition, our self-distillation integrates the knowledge of both streams to generate pseudo-targets, which can also be regarded as a kind of interaction.

## 3 Method

In this section, we introduce our TwoStream-SLR and TwoStream-SLT for the SLR and SLT tasks, respectively. Given a sign video $\mathcal{V} = (v_1, ..., v_T)$ with $T$ frames, our goal is to optimize TwoStream-SLR that can predict the gloss sequence $\mathcal{G} = (g_1, ..., g_U)$ with $U$ glosses (*i.e.*, SLR). Appending a translation network onto TwoStream-SLR yields TwoStream-SLT, a model which is capable of predicting associated spoken language sentence $\mathcal{S} = (s_1, ..., s_W)$ with $W$ words (*i.e.*, SLT) from sign videos. In general, $U \neq W$. The proposed TwoStream-SLR and TwoStream-SLT are introduced in Section 3.1 and Section 3.2, respectively.

### 3.1 TwoStream-SLR

Figure 2 shows the overview of our TwoStream-SLR, which consists of five parts: 1) a video encoder; 2) a keypoint sequence encoder; 3) a joint head; 4) a bidirectional lateral connection module; 5) two sign pyramid networks with auxiliary supervision, to model RGB videos and keypoint sequences.

**Video Encoder.** We use S3D [51] with a lightweight head network as our video encoder. Only the first four blocks of S3D are used since our goal is to extract dense representations along the temporal

dimension. We feed each $T \times H \times W \times 3$ video into the encoder to extract its features, where $T$ denotes the frame number, $H$ and $W$ represent the height and width of the sign video. We set $H$ and $W$ as 224 by default. The output feature of the last S3D block is spatially pooled into the size of $T/4 \times 832$ before it is fed into the head network. The goal of the head network is to further capture the temporal context. It consists of a temporal linear layer, a batch normalization layer, a ReLU layer, as well as a temporal convolutional block which contains two temporal convolutional layers with a temporal kernel size of 3 and a stride of 1, a linear translation layer, and a ReLU layer. The output feature which is named as gloss representation has a size of $T/4 \times 512$. Then a linear classifier and a Softmax function are applied to extract frame-level gloss probabilities. At last, we use connectionist temporal classification (CTC) loss $\mathcal{L}_{CTC}^{V}$ to optimize the video encoder.

**Keypoint Encoder.** To model keypoint sequences, we first utilize the HRNet [50] which is trained on COCO-WholeBody [23] to generate 42 hand keypoints, 68 face keypoints covering the mouth, eyes, and face contour, and 11 upper body keypoints covering shoulders, elbows, and wrists per frame. We empirically found that using only a subset of 26 face keypoints (10 mouth keypoints and 16 for other parts) performs well while saving computational resources. In total, 79 keypoints are used. We use heatmaps to represent the keypoints. Concretely, denoting the keypoint heatmap sequence with a size of $T \times H' \times W' \times K$ as $\mathbf{G}$, where $H'$ and $W'$ represent the spatial resolution of each heatmap, and $K$ is the total number of keypoints, the elements of $\mathbf{G}$ are generated by a Gaussian function: $\mathbf{G}_{(t,i,j,k)} = \exp\left(-[(i - x_t^k)^2 + (j - y_t^k)^2]/2\sigma^2\right)$, where $(x_t^k, y_t^k)$ denotes the coordinates of the $k$-th keypoint in the $t$-th frame, and $\sigma$ controls the scale of keypoints. We set $\sigma = 4$ by default and $H' = W' = 112$ to reduce computational cost. The network architecture of the keypoint encoder is the same as the video encoder, except for the first convolutional layer which is substituted to fit the keypoint input. Note that weights of the video encoder and keypoint encoder are not shared. Similarly, a CTC loss is utilized to train the keypoint encoder, which is denoted as $\mathcal{L}_{CTC}^{K}$.

**Bidirectional Lateral Connection.** To fuse the information of two streams, we propose lateral connection, which is explored in action recognition [17, 12, 16, 15] and object detection [33]. The lateral connection is implemented as an element-wise add operation between two feature maps of the same resolution. We apply lateral connection on the features ($C_1$, $C_2$, and $C_3$) generated by the first three blocks ($B_1$, $B_2$, and $B_3$) of the two S3D backbones. Since the spatial resolutions of intermediate features extracted by the two streams are different, we use spatially strided convolution and transposed convolution to align their spatial resolutions. For implementation, the lateral connection is bidirectional, and other variants such as unidirectional lateral connection are studied in Section 4.2.

**Joint Head and Late Ensemble.** Both the video encoder and keypoint encoder have their own head networks. To fully explore the potential of our dual encoder architecture, we present an additional head network named joint head, which takes the concatenation of outputs of the two S3D networks as inputs. The architecture of the joint head is the same as the video head and keypoint head. The joint head is supervised by a CTC loss as well, which is denoted as $\mathcal{L}_{CTC}^{J}$. We average the frame-wise gloss probabilities predicted by the video head, keypoint head, and joint head and feed them to a CTC decoder to generate the gloss sequence $\mathcal{G}$. This late ensemble strategy fuses multi-stream results and improves over single-stream predictions as shown in our experiments.

**Sign Pyramid Network.** To better capture glosses of different temporal spans and efficiently supervise the shallow layers to learn meaningful representations, we build a sign pyramid network (SPN) with auxiliary supervision upon the dual visual encoder. The architecture of SPN is illustrated in Figure 3a. Specifically, we denote outputs of the last three blocks of the S3D backbone as $C_2$, $C_3$, and $C_4$, respectively. Similar to [52], the construction of the sign pyramid involves a top-down pathway and a lateral connection. We use element-wise add operation to fuse features extracted by adjacent S3D blocks, and the fused features are termed as $P_2$ and $P_3$ (see Figure 3a). We use transposed convolution to match both temporal and spatial dimensions of two feature maps before element-wise addition. Then two separate head networks with the same architecture as the one used in the dual encoder are applied on $P_2$ and $P_3$ to extract frame-level gloss probabilities. Similarly, CTC losses are adopted to provide auxiliary supervision. Without loss of generality, we use two independent SPNs for the video and keypoint stream. The auxiliary CTC losses of two streams are denoted as $\mathcal{L}_{ACTC}^{V}$ and $\mathcal{L}_{ACTC}^{K}$, respectively. SPN is dropped in the inference stage.

**Frame-Level Self-Distillation.** Existing datasets only provide sentence-level gloss annotations, where gloss temporal boundaries are not labeled. Thus, CTC [19] loss is widely adopted to leverage this kind of weak supervision. However, once well optimized, the visual encoder is able to generate

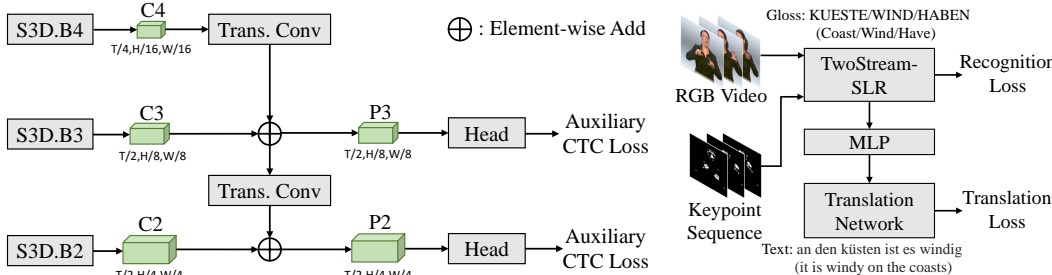

(a) Architecture of sign pyramid network (SPN).      (b) Overview of TwoStream-SLT.

Figure 3: (a) The sign pyramid network takes the features ($C_2$, $C_3$, and $C_4$) of the last three blocks of S3D backbone as inputs and generate the fused features $P_2$ and $P_3$. The construction of the sign pyramid involves a top-down pathway and a lateral connection where the transposed convolution is used to match both temporal and spatial dimensions of two feature maps. Two separate head networks of the same architecture are applied on $P_2$ and $P_3$ to generate frame-level gloss probabilities. We use the CTC loss as auxiliary supervision. Each stream in our dual encoder has an independent SPN. (b) Simply appending a translation network onto our TwoStream-SLR yields TwoStream-SLT, a framework for SLT. TwoStream-SLT is jointly supervised by the recognition loss and the translation loss.

frame-wise gloss probabilities from which one can estimate the approximate temporal boundaries of glosses. This inspires us to use the predicted frame-wise gloss probabilities as pseudo-targets to provide extra fine-grained supervision in addition to the coarse-grained CTC loss. In accordance with our two-stream design, we use the averaged gloss probabilities from the three head networks as pseudo-targets to guide the learning of each stream. Formally, we minimize the KL-divergence between the pseudo-targets and the predictions of the three head networks. We call it frame-level self-distillation loss $\mathcal{L}_{Dist}$ as it not only provides frame-level supervision but also distills knowledge in the late ensemble back into each individual stream.

**Loss Function.** The overall loss of TwoStream-SLR is composed of three parts: 1) the CTC losses applied on the outputs of the video encoder ($\mathcal{L}_{CTC}^V$), keypoint encoder ($\mathcal{L}_{CTC}^K$), and joint head ($\mathcal{L}_{CTC}^J$); 2) the auxiliary CTC losses ($\mathcal{L}_{ACTC}^V$ and $\mathcal{L}_{ACTC}^K$) applied on the outputs of two sign pyramid networks; 3) the distillation loss ($\mathcal{L}_{Dist}$). We formulate the recognition loss as follows:

$$\mathcal{L}_{SLR} = \mathcal{L}_{CTC}^V + \mathcal{L}_{CTC}^K + \mathcal{L}_{CTC}^J + \lambda_V \mathcal{L}_{ACTC}^V + \lambda_K \mathcal{L}_{ACTC}^K + \mathcal{L}_{Dist}, \tag{1}$$

where $\lambda_V$ and $\lambda_K$ denote loss weights of the auxiliary CTC loss of the video stream and keypoint stream. Up to now, we have introduced all components of TwoStream-SLR. Once the training is finished, TwoStream-SLR is capable of predicting a gloss sequence by averaging predictions from the three head networks.

### 3.2 TwoStream-SLT

Previous approaches [6, 8, 31, 56, 55, 10] formulate the SLT task as a neural machine translation (NMT) problem, where the input of the translation network is the output of the visual encoder. We follow this paradigm and append an MLP with two hidden layers and a subsequent translation network onto TwoStream-SLR to enable SLT. The resulting network is called TwoStream-SLT, which is illustrated in Figure 3b. We adopt mBART [34] as our translation network due to its excellent SLT performance [10]. To take advantage of our TwoStream design, we append an MLP and a translation network to each of the three heads of our TwoStream-SLR. The inputs of each MLP are the features (*i.e.*, gloss representations defined in Section 3.1) encoded by the corresponding head network. The translation loss is a standard sequence-to-sequence cross-entropy loss [49]. The overall translation loss $\mathcal{L}_T$ is the sum of three translation losses. TwoStream-SLT is jointly supervised by the recognition loss $\mathcal{L}_{SLR}$ defined in Eq. 1 and the translation loss $\mathcal{L}_T$, which can be formulated as:

$$\mathcal{L}_{SLT} = \mathcal{L}_{SLR} + \mathcal{L}_T. \tag{2}$$

During inference, we adopt the fusion strategy for multi-source translation ensemble [18] to combine predictions of the three translation networks. More details can be found in the supplementary materials.

Table 1: Comparison with previous works on **Sign Language Recognition (SLR)**. WER is adopted as the evaluation metric. Previous best results are underlined. $*$ denotes methods which adopt other modalities besides RGB videos such as pose keypoints [56], optical flow [14], or hand/mouth shape [30]. The results of [5, 21, 14, 11, 8] on CSL-Daily are reproduced by SignBT [55].

| Method | Phoenix-2014 | | Phoenix-2014T | | CSL-Daily | |
|---|---|---|---|---|---|---|
| | Dev | Test | Dev | Test | Dev | Test |
| SubUNets [5] | 40.8 | 40.7 | - | - | 41.4 | 41.0 |
| LS-HAN [21] | - | - | - | - | 39.0 | 39.4 |
| IAN [40] | 37.1 | 36.7 | - | - | - | - |
| ReSign [28] | 27.1 | 26.8 | - | - | - | - |
| CNN-LSTM-HMMs (Multi-Stream)$*$ [30] | 26.0 | 26.0 | 22.1 | 24.1 | - | - |
| SFL [37] | 24.9 | 25.3 | 25.1 | 26.1 | - | - |
| DNF (RGB) [14] | 23.8 | 24.4 | - | - | 32.8 | 32.4 |
| FCN [11] | 23.7 | 23.9 | 23.3 | 25.1 | 33.2 | 33.5 |
| DNF (RGB+Flow)$*$ [14] | 23.1 | 22.9 | - | - | - | - |
| Joint-SLRT [8] | - | - | 24.6 | 24.5 | 33.1 | 32.0 |
| VAC [36] | 21.2 | 22.3 | - | - | - | - |
| LCSA [59] | 21.4 | 21.9 | - | - | - | - |
| CMA [41] | 21.3 | 21.9 | - | - | - | - |
| SignBT [55] | - | - | 22.7 | 23.9 | 33.2 | 33.2 |
| MMTLB [10] | - | - | 21.9 | 22.5 | - | - |
| SMKD [20] | 20.8 | 21.0 | 20.8 | 22.4 | - | - |
| STMC-R (RGB+Pose)$*$ [56] | 21.1 | 20.7 | 19.6 | 21.0 | - | - |
| $C^2$SLR (RGB+Pose)$*$ [58] | 20.5 | 20.4 | 20.2 | 20.4 | - | - |
| TwoStream-SLR (Ours)$*$ | **18.4** | **18.8** | **17.7** | **19.3** | **25.4** | **25.3** |

## 4 Experiment

**Implementation Details.** The S3D backbone is first pretrained on Kinetics-400 [24]. Then we separately pretrain the video and keypoint encoder without the sign pyramid network under the supervision of a single CTC loss. Finally, TwoStream-SLR loads the pretrained weights of both encoders for SLR training with loss defined in Eq. 1. Following [10], we initialize our translation network with mBART-large-cc25[2] pretrained on CC25[3] and freeze the S3D backbones during SLT training to prevent overfitting. Unless otherwise specified, we set $\lambda_V = 0.2$ and $\lambda_K = 0.5$ in Eq. 1, beam width as 5 for the CTC decoder and the SLT decoder during inference. We use cosine annealing schedule of 40 epochs and an Adam optimizer with weight decay $1e - 3$, initial learning rate $1e - 3$ for TwoStream-SLR and $1e - 5$ for the MLP and translation network in TwoStream-SLT. We train our models on 8 Nvidia V100 GPUs. Please refer to the supplementary materials for more details.

**Datasets.** We use Phoenix-2014 [27], Phoenix-2014T [6], and CSL-Daily [55] to evaluate our method on SLR, while the last two datasets are also leveraged for SLT evaluation since they provide text annotations. All ablation studies are conducted on the Phoenix-2014T SLR task.

- **Phoenix-2014** is a German SLR dataset with a vocabulary size of 1081 for glosses. It consists of 5672, 540, and 629 samples in the training, dev, and test set.

- **Phoenix-2014T** is an extension of Phoenix-2014 with a vocabulary size of 1066 for glosses and 2887 for German text. There are 7096, 519, and 642 samples in the training, dev, and test set.

- **CSL-Daily** is a newly released large-scale Chinese sign language dataset with a vocabulary size of 2000 for glosses and 2343 for Chinese text. It consists of 18401, 1077, and 1176 samples in the training, dev, and test set.

**Evaluation Metrics.** Following previous works [56, 10, 8, 6, 55], we adopt word error rate (WER) for SLR evaluation, and BLEU [39] and ROUGE-L [32] to evaluate SLT. Lower WER indicates better recognition performance. For BLEU and ROUGE-L, the higher, the better.

---

[2]https://huggingface.co/facebook/mbart-large-cc25
[3]https://commoncrawl.org/

Table 2: Comparison with previous works on **Sign Language Translation (SLT)**. Sign2Gloss2Text indicates a two-staged pipeline and Sign2Text indicates end-to-end sign-to-text translation. Previous best results are underlined. † denotes methods without using gloss annotations. (R: ROUGE, B: BLEU.) The results of [6, 8] on CSL-Daily are reproduced by SignBT [55].

| | Phoenix-2014T | | | | | | | | | |
|---|---|---|---|---|---|---|---|---|---|---|
| | | | Dev | | | | | Test | | |
| Sign2Gloss2Text | R | B1 | B2 | B3 | B4 | R | B1 | B2 | B3 | B4 |
| SL-Luong [6] | 44.14 | 42.88 | 30.30 | 23.02 | 18.40 | 43.80 | 43.29 | 30.39 | 22.82 | 18.13 |
| Joint-SLRT [8] | - | 47.73 | 34.82 | 27.11 | 22.11 | - | 48.47 | 35.35 | 27.57 | 22.45 |
| SignBT [55] | 49.53 | 49.33 | 36.43 | 28.66 | 23.51 | 49.35 | 48.55 | 36.13 | 28.47 | 23.51 |
| STMC-Transf [53] | 46.31 | 48.27 | 35.20 | 27.47 | 22.47 | 46.77 | 48.73 | 36.53 | 29.03 | 24.00 |
| MMTLB [10] | 50.23 | 50.36 | 37.50 | 29.69 | 24.63 | 49.59 | 49.94 | 37.28 | 29.67 | 24.60 |
| TwoStream-SLT (Ours) | **52.01** | **52.35** | **39.76** | **31.85** | **26.47** | **51.59** | **52.11** | **39.81** | **32.00** | **26.71** |
| Sign2Text | R | B1 | B2 | B3 | B4 | R | B1 | B2 | B3 | B4 |
| SL-Luong† [6] | 31.80 | 31.87 | 19.11 | 13.16 | 9.94 | 31.80 | 32.24 | 19.03 | 12.83 | 9.58 |
| TSPNet† [31] | - | - | - | - | - | 34.96 | 36.10 | 23.12 | 16.88 | 13.41 |
| Joint-SLRT [8] | - | 47.26 | 34.40 | 27.05 | 22.38 | - | 46.61 | 33.73 | 26.19 | 21.32 |
| STMC-T [56] | 48.24 | 47.60 | 36.43 | 29.18 | 24.09 | 46.65 | 46.98 | 36.09 | 28.70 | 23.65 |
| SignBT [55] | 50.29 | 51.11 | 37.90 | 29.80 | 24.45 | 49.54 | 50.80 | 37.75 | 29.72 | 24.32 |
| MMTLB [10] | 53.10 | 53.95 | 41.12 | 33.14 | 27.61 | 52.65 | 53.97 | 41.75 | 33.84 | 28.39 |
| TwoStream-SLT (Ours) | **54.08** | **54.32** | **41.99** | **34.15** | **28.66** | **53.48** | **54.90** | **42.43** | **34.46** | **28.95** |

| | CSL-Daily | | | | | | | | | |
|---|---|---|---|---|---|---|---|---|---|---|
| | | | Dev | | | | | Test | | |
| Sign2Gloss2Text | R | B1 | B2 | B3 | B4 | R | B1 | B2 | B3 | B4 |
| SL-Luong [6] | 40.18 | 41.46 | 25.71 | 16.57 | 11.06 | 40.05 | 41.55 | 25.73 | 16.54 | 11.03 |
| Joint-SLRT [8] | 44.18 | 46.82 | 32.22 | 22.49 | 15.94 | 44.81 | 47.09 | 32.49 | 22.61 | 16.24 |
| SignBT [55] | 48.38 | 50.97 | 36.16 | 26.26 | 19.53 | 48.21 | 50.68 | 36.00 | 26.20 | 19.67 |
| MMTLB [10] | 51.35 | 50.89 | 37.96 | 28.53 | 21.88 | 51.43 | 50.33 | 37.44 | 28.08 | 21.46 |
| TwoStream-SLT (Ours) | **53.91** | **53.58** | **40.49** | **30.67** | **23.71** | **54.92** | **54.08** | **41.02** | **31.18** | **24.13** |
| Sign2Text | R | B1 | B2 | B3 | B4 | R | B1 | B2 | B3 | B4 |
| SL-Luong† [6] | 34.28 | 34.22 | 19.72 | 12.24 | 7.96 | 34.54 | 34.16 | 19.57 | 11.84 | 7.56 |
| Joint-SLRT [8] | 37.06 | 37.47 | 24.67 | 16.86 | 11.88 | 36.74 | 37.38 | 24.36 | 16.55 | 11.79 |
| SignBT [55] | 49.49 | 51.46 | 37.23 | 27.51 | 20.80 | 49.31 | 51.42 | 37.26 | 27.76 | 21.34 |
| MMTLB [10] | 53.38 | 53.81 | 40.84 | 31.29 | 24.42 | 53.25 | 53.31 | 40.41 | 30.87 | 23.92 |
| TwoStream-SLT (Ours) | **55.10** | **55.21** | **42.31** | **32.71** | **25.76** | **55.72** | **55.44** | **42.59** | **32.87** | **25.79** |

## 4.1 Comparison with State-of-the-art Methods on SLR and SLT

For SLR, we compare our TwoStream-SLR with state-of-the-art methods on Phoenix-2014, Phoenix-2014T, and CSL-Daily, as shown in Table 1. Our recognition network achieves a new state-of-the-art on all datasets and outperforms the previous best method by 1.6% on Phoenix-2014, 1.1% on Phoenix-2014T, and 6.7% on CSL-Daily, respectively. For SLT, we compare our TwoStream-SLT with state-of-the-art methods on Phoenix-2014T and CSL-Daily as shown in Table 2. Following common practice, we evaluate our method in two settings: 1) Sign2Text (as described in Section 3.2) which directly generates texts given sign videos; 2) Sign2Gloss2Text where we first use a recognition model to predict gloss sequences from sign videos, and then a translation model trained on gloss-text pairs to translate predicted gloss sequences to texts. Our TwoStream-SLT surpasses all previous methods in both settings. We believe that our better SLT performance is mainly attributed to the superior visual representations encoded by TwoStream-SLR.

## 4.2 Ablation Study

**Effects of Each Component.** We first show the effects of each proposed component of TwoStream-SLR in Table 3. Without the dual architecture, the two single streams (one models RGB videos and the other models keypoint sequences) achieve 21.08% and 27.14% WER on the Phoenix-2014T Dev set. By averaging the final predictions from two streams, the WER is reduced to 20.47%. The proposed bidirectional lateral connection brings in information interaction between two streams, leading to the significant improvement of 1.44%. Introducing the sign pyramid network (SPN) with auxiliary

Table 3: Study the effects of each component of TwoStream-SLR on the Phoenix-2014T SLR task. (V: video, K: keypoint, Bilateral: bidirectional lateral connection, SPN: sign pyramid network.)

| V-Encoder | K-Encoder | Bilateral | SPN | Joint Head | Distillation | Dev | Test |
|:---:|:---:|:---:|:---:|:---:|:---:|:---:|:---:|
| ✓ | | | | | | 21.08 | 22.42 |
| | ✓ | | | | | 27.14 | 27.19 |
| ✓ | ✓ | | | | | 20.47 | 21.55 |
| ✓ | ✓ | ✓ | | | | 19.03 | 20.12 |
| ✓ | ✓ | ✓ | ✓ | | | 18.52 | 19.91 |
| ✓ | ✓ | ✓ | ✓ | ✓ | | 18.36 | 19.49 |
| ✓ | ✓ | ✓ | ✓ | ✓ | ✓ | **17.72** | **19.32** |

Table 4: Ablation studies of: (a) lateral connection; (b) sign pyramid network (SPN); (c) weights of the auxiliary CTC losses; (d) the weight of the distillation loss; (e) self-distillation strategies, on the Phoenix-2014T SLR task. See Section 3.1 for the definition of $C_i$ and $P_i$. (V: video stream, K: keypoint stream.)

| V→K | K→V | Connection | Dev | Test |
|:---:|:---:|:---:|:---:|:---:|
| | | None | 18.57 | 20.03 |
| ✓ | | $C_1, C_2, C_3$ | 17.88 | 19.61 |
| | ✓ | $C_1, C_2, C_3$ | 18.82 | 19.93 |
| ✓ | ✓ | $C_1, C_2, C_3$ | **17.72** | **19.32** |
| ✓ | ✓ | $C_2, C_3$ | 17.91 | 19.54 |

(a) Lateral connection.

| SPN-V | SPN-K | Level | Dev | Test |
|:---:|:---:|:---:|:---:|:---:|
| ✓ | | $P_2, P_3$ | 17.99 | 19.39 |
| | ✓ | $P_2, P_3$ | 18.15 | 19.42 |
| ✓ | ✓ | $P_2, P_3$ | **17.72** | **19.32** |
| ✓ | ✓ | $P_1, P_2, P_3$ | 18.07 | 19.35 |
| ✓ | ✓ | $P_3$ | 17.96 | 19.51 |

(b) Sign pyramid network (SPN).

| $\lambda_V$ | $\lambda_K$ | Dev | Test |
|:---:|:---:|:---:|:---:|
| 0.1 | 0.5 | 17.85 | 19.42 |
| 0.2 | 0.5 | **17.72** | 19.32 |
| 0.5 | 0.5 | 17.99 | 19.30 |
| 0.2 | 0.1 | 17.80 | **19.14** |
| 0.2 | 0.2 | **17.72** | 19.49 |

(c) Weights of the auxiliary CTC loss.

| Weight of $\mathcal{L}_{Dist}$ | Dev | Test |
|:---:|:---:|:---:|
| 0.2 | 18.41 | **19.21** |
| 0.5 | 18.20 | 19.63 |
| 1.0 | **17.72** | 19.32 |
| 1.5 | 18.28 | 19.93 |
| 2.0 | 17.83 | 19.28 |

(d) The weight of the distillation loss.

| Teacher | Students | Target | Dev | Test |
|:---:|:---:|:---:|:---:|:---:|
| Joint Head | V, K | Soft | 18.82 | 19.93 |
| Ensemble | V, K | Soft | 18.12 | 19.68 |
| Ensemble | V, K, J | Soft | **17.72** | **19.32** |
| Ensemble | V, K, J | Hard | 18.25 | 19.58 |

(e) Distillation strategies. "J" denotes joint head.

CTC losses facilitates intermediate layers to learn more meaningful features, further boosting the performance to 18.52%. As described in Section 3.1, besides the individual head networks, we present a joint head to further integrate the encoded features from two streams. Our approach equipped with the joint head achieves 18.36 WER. At last, by adding the auxiliary frame-wise self-distillation loss, our framework attains the best result, yielding the WER of 17.72.

**Study on Lateral Connection.** The goal of lateral connection is to provide information interaction such that the video stream and keypoint stream can complement each other. Here we compare our default configuration where the bidirectional lateral connection is conducted on $C_1$, $C_2$, and $C_3$ (see Section 3.1 for their definitions) of two streams, with other variants including unidirectional lateral connection and different connection strategies. The comparison is shown in Table 4a. Without the lateral connection, the baseline model achieves 18.57 WER on the Phoenix-2014T Dev set. Thanks to the information interaction, both unidirectional (video→keypoint and keypoint→video) and bidirectional lateral connection strategies outperform the baseline. We adopt the bidirectional lateral connection performed on $C_1$, $C_2$, and $C_3$ due to its best performance.

**Study on Sign Pyramid Network and Auxiliary Supervisions.** Sign language understanding suffers from data scarcity. To capture glosses of various temporal spans and drive intermediate layers to learn more meaningful features, we propose a sign pyramid network (SPN) with auxiliary CTC losses. Here we study three key factors: 1) applying SPN on a single stream or both streams; 2) the levels of SPN; 3) the loss weights of auxiliary CTC losses of the two streams. The first two factors are studied in Table 4b while the last one in Table 4c. We observe that applying SPN on both video and keypoint

Table 5: Ablation studies of: (a) various combinations of keypoints as the inputs of our keypoint encoder; (b) the keypoint scale $\sigma$ of the Gaussian function and the resolution of the generated heatmaps, on the Phoenix-2014T SLR task.

| Upper body | Hand | Mouth | Face | #Keypoints | Dev | Test |
|:---:|:---:|:---:|:---:|:---:|:---:|:---:|
| ✓ | | | | 11 | 49.11 | 48.46 |
| ✓ | ✓ | | | 53 (+42) | 37.15 | 36.88 |
| ✓ | ✓ | ✓ | | 63 (+10) | 28.42 | 28.20 |
| ✓ | ✓ | ✓ | ✓ | 79 (+16) | **27.14** | **27.19** |

| $\sigma$ | $(H', W')$ | Dev | Test |
|:---:|:---:|:---:|:---:|
| 1 | 56 | 29.78 | 28.72 |
| 2 | 56 | 30.10 | 29.02 |
| 2 | 112 | 27.22 | 27.52 |
| 4 | 112 | **27.14** | 27.19 |
| 6 | 112 | 27.94 | **27.10** |

(a) The effects of different combinations of keypoints as inputs of the keypoint encoder.

(b) Keypoint scale $\sigma$ and heatmap resolution.

streams yields better results. We also find imposing extra CTC supervision on very shallow layers (*i.e.*, $P_1$) hurts the performance, thus we advocate generating two levels of pyramid ($P_2$ and $P_3$) for each stream. We set $\lambda_V$ and $\lambda_K$ in Eq. 1 as 0.2 and 0.5 as they give better performance.

**Study on Self-Distillation Strategies.** The proposed self-distillation loss provides frame-level supervision. There is a trade-off between the pseudo fine-grained supervision (self-distillation loss) and the coarse-grained supervision (CTC loss). Here we vary the weight of self-distillation loss $\mathcal{L}_{Dist}$ and show the results in Table 4d. The best performance is obtained when the weight is set to 1.0. Besides, we also study: 1) which prediction should be the pseudo-target (teacher) 2) which heads in our dual visual encoder should be taught by the pseudo-target (students); 3) whether to binarize the probabilities of the pseudo-target (soft target or hard target). Table 4e shows the results. As described in Section 3.1, we present three head networks in the dual visual encoder, namely the video head, keypoint head, and joint head. We observe that using averaged gloss probabilities from three heads (Ensemble) as the pseudo-target outperforms using predictions of the joint head. We also find that applying the self-distillation loss on all three heads achieves better results than teaching only two heads, and using soft predictions as pseudo-targets outperforms the one with hard pseudo-targets.

**Study on Keypoint Inputs.** Sign languages utilize multiple visual signals including handshape, facial expressions, the movement of body, head, mouth, and eyes, to convey information. To investigate the importance of various keypoints in SLR, we train several single-stream keypoint encoders using different combinations of keypoints as inputs and evaluate their performance on the Phoenix-2014T SLR task. We use the HRNet trained on COCO-WholeBody to generate 79 keypoints in total. These keypoints can be divided into 4 groups: 1) 11 upper body keypoints; 2) 42 hand keypoints; 3) 10 mouth keypoints; 4) 16 face keypoints (excluding the mouth). Step by step, we add each group of keypoints into model training and the results are shown in Table 5a. It can be seen that all parts contribute to sign language understanding. As described in Section 3.1, we use a Gaussian function with a hyper-parameter $\sigma$, which denotes keypoint scale, to generate a set of heatmaps, each of size $H' \times W'$ to represent keypoint sequences. Here we study the value of $\sigma$ and the resolution of heatmaps in Table 5b. We find that $\sigma = 4$ and $H' = W' = 112$ achieves the best performance.

## 5 Conclusion

In this paper, we concentrate on how to introduce domain knowledge into sign language understanding. To achieve the goal, we present a novel framework named TwoStream-SLR which adopts two streams to model RGB videos and keypoint sequences for sign language recognition. A variety of techniques are proposed to make the two streams interact with each other, including bidirectional lateral connection, sign pyramid network, and frame-level self-distillation. We further extend TwoStream-SLR to a sign language translation model by attaching an MLP and a translation network, yielding the translation framework named TwoStream-SLT. Our TwoStream-SLR and TwoStream-SLT achieve state-of-the-art performance on SLR and SLT tasks across a series of datasets including Phoenix-2014, Phoenix-2014T, and CSL-Daily. We hope that our approach can serve as a baseline to facilitate future research.

**Acknowledgements.** The work described in this paper was partially supported by a grant from the Research Grants Council of the HKSAR, China (Project No. HKUST16200118).

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
