# Two-Stream Network for Sign Language Recognition and Translation

## A   Loss Formulation

**CTC Loss.** We apply the CTC loss [3] not only on the outputs of the video head, pose head, and joint head, but also on the outputs of the sign pyramid networks for auxiliary training. The CTC loss considers all feasible alignments between two sequences. Specifically, given a video $\mathcal{V}$ containing $T$ frames and its annotated gloss sequence $\mathcal{G}$ containing $U$ glosses, CTC yields $p(\mathcal{G}|\mathcal{V})$ by marginalizing over all feasible decoding paths:

$$p(\mathcal{G}|\mathcal{V}) = \sum_{\pi \in \mathcal{S}} p(\pi|\mathcal{V}), \tag{1}$$

where $\pi$ denotes a frame-level gloss path of length $T/4$ in our model and $\mathcal{S}$ is the set of all feasible mappings between $\mathcal{V}$ and $\mathcal{G}$. The probability $p(\pi|\mathcal{V})$ is computed by applying the Softmax function on the outputs of each head network. Finally, the CTC loss is defined as:

$$\mathcal{L}_{CTC} = -\ln p(\mathcal{G}|\mathcal{V}). \tag{2}$$

**Self-Distillation Loss.** Given a sign video $\mathcal{V}$, we use $p_i^v$, $p_i^k$, and $p_i^j$ to denote its probability at timestamp $i$ generated by the video head, keypoint head, and joint head, respectively. The averaged prediction $p_i^a$ serves as the pseudo-target and it is calculated by:

$$p_i^a = \frac{1}{3}\left(p_i^v + p_i^k + p_i^j\right). \tag{3}$$

Then the self-distillation loss $\mathcal{L}_{Dist}$ can be formulated as:

$$\mathcal{L}_{Dist} = \sum_{i=1}^{T/4} \left( D(p_i^a \parallel p_i^v) + D(p_i^a \parallel p_i^k) + D(p_i^a \parallel p_i^j) \right), \tag{4}$$

where $D(x \parallel y)$ represents the KL-divergence between the probability distributions $x$ and $y$, and $T/4$ is the length of the output.

**Translation Loss.** Given a sign video $\mathcal{V}$ and its associated spoken language sentence $\mathcal{S} = (s_1, ..., s_W)$ with $W$ words, the translation loss $\mathcal{L}_T$ is a sequence-to-sequence cross-entropy loss defined as:

$$\mathcal{L}_T = -\ln p(\mathcal{S}|\mathcal{V}) = -\sum_{i=1}^{W} \ln p(s_i|s_{<i}, \mathcal{V}). \tag{5}$$

## B   More Implementation Details

### B.1   TwoStream-SLR

**Training.** The S3D backbone[1] is pretrained on Kinetics-400 [4] by [5]. We separately train the video encoder and the keypoint encoder without the sign pyramid network via a single CTC loss. Then

---

[1]https://github.com/kylemin/S3D

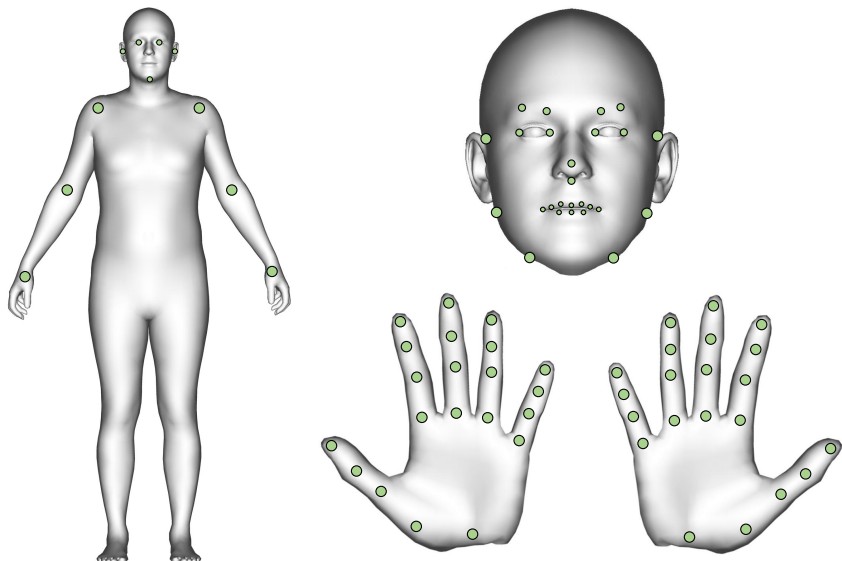

Figure 1: Illustration of keypoints used in our approach.

the weights of two pretrained encoders are loaded into our TwoStream-SLR for SLR training. The newly added components including the sign pyramid networks and joint head are randomly initialized. Data augmentations include spatial cropping in the range of [0.7-1.0] and frame-rate augmentation in the range of [$\times 0.5$-$\times 1.5$]. We adopt identical data augmentations for RGB videos and heatmap sequences to maintain spatial and temporal consistency. We use the first four blocks of the S3D backbone and freeze the first block during training. For each training stage, we train our model on 8 Nvidia V100 GPUs for 40 epochs with the initial learning rate $1e-3$, a cosine annealing schedule, and an Adam optimizer with weight decay $1e-3$.

**Inference.** We drop the sign pyramid networks in the inference stage. A CTC decoder is adopted to yield the final gloss predictions. Specifically, denoting the gloss probabilities predicted by the video head, keypoint head, and joint head at the $i$-th frame as $p_i^v, p_i^k$, and $p_i^j$, respectively, we first compute the average probability $p_i^a = (p_i^v + p_i^k + p_i^j)/3$. Then the CTC decoder takes $\{p_i^a\}_{i=1}^{T/4}$ as inputs and outputs the final gloss prediction using the beam search algorithm with a beam width of 5. More details are available in [3].

### B.2 TwoStream-SLT

**Training.** We attach an MLP and a translation network onto each of the three heads in our TwoStream-SLR model to perform SLT. The resulting network is termed as TwoStream-SLT. Following [1], we use mBART-large-cc25[2] as our translation network. Before SLT training, we pretrain our translation networks on gloss-text pairs. All text sequences are tokenized into subword tokens by the sentence-piece model used in mBART. A special language id token, *i.e.* <de_DE> for German and <zh_CN> for Chinese, is prepended to the text sequence to identify the target language. During SLT training, we freeze the S3D backbones and tune the video head, keypoint head, joint head, MLPs, and translation networks. We experimentally find that this strategy performs better than training the whole network while reducing computational cost. We train our TwoStream-SLT for 40 epochs, using a cosine annealing schedule and an Adam optimizer with weight decay $1e-3$. The initial learning rate is set to $1e-3$ for the visual heads and $1e-5$ for the MLPs and translation networks.

**Inference.** We adopt the fusion strategy for multi-source translation ensemble proposed in [2] to aggregate results of the three translation networks. Concretely, given a sign video $\mathcal{V}$ and tokens $s_{<i}$ before step $i$, we average the predictions from the three independent translation networks to obtain

---

[2]https://huggingface.co/facebook/mbart-large-cc25

Table 1: Ablation studies of **Sign Language Translation (SLT)** on Phoenix-2014T and CSL-Daily. SingleStream-SLT which only utilizes a single video encoder without modelling keypoints serves as our baseline. TwoStream-SLT-V/K/J denotes the network where only one translation network is attached onto the video head/keypoint head/joint head. TwoStream-SLT-V*/K*/J* denotes that we train TwoStream-SLT-V/K/J with three different random seeds, and predictions of the three independently trained models attached to the same head are averaged to generate final results.

| | Phoenix-2014T | | | | | | | | | |
| | | Dev | | | | | Test | | | |
| Framework | R | B1 | B2 | B3 | B4 | R | B1 | B2 | B3 | B4 |
|---|---|---|---|---|---|---|---|---|---|---|
| SingleStream-SLT | 52.60 | 53.35 | 40.75 | 32.92 | 27.59 | 52.15 | 53.85 | 41.33 | 33.53 | 28.16 |
| TwoStream-SLT-V | 53.11 | 53.62 | 41.02 | 33.02 | 27.53 | 53.28 | 54.38 | 42.04 | 34.14 | 28.68 |
| TwoStream-SLT-K | 53.21 | 53.88 | 41.14 | 33.26 | 27.83 | 52.87 | 54.58 | 41.78 | 33.60 | 27.98 |
| TwoStream-SLT-J | 53.05 | 53.94 | 41.37 | 33.50 | 28.11 | 52.74 | 54.00 | 41.66 | 33.72 | 28.23 |
| TwoStream-SLT-V* | 53.89 | 54.05 | 41.68 | 33.69 | 28.21 | 53.27 | 54.57 | 42.24 | 34.25 | 28.70 |
| TwoStream-SLT-K* | 53.32 | 53.66 | 41.31 | 33.55 | 28.10 | 53.19 | 54.22 | 41.72 | 33.82 | 28.42 |
| TwoStream-SLT-J* | 53.51 | 54.03 | 41.51 | 33.51 | 27.97 | 53.17 | 54.11 | 41.78 | 33.88 | 28.38 |
| TwoStream-SLT | **54.08** | **54.32** | **41.99** | **34.15** | **28.66** | **53.48** | **54.90** | **42.43** | **34.46** | **28.95** |

| | CSL-Daily | | | | | | | | | |
| | | Dev | | | | | Test | | | |
| Framework | R | B1 | B2 | B3 | B4 | R | B1 | B2 | B3 | B4 |
|---|---|---|---|---|---|---|---|---|---|---|
| SingleStream-SLT | 53.33 | 53.29 | 40.37 | 30.81 | 23.95 | 53.32 | 52.84 | 40.12 | 30.57 | 23.70 |
| TwoStream-SLT-V | 54.50 | 54.22 | 41.46 | 31.87 | 24.93 | 54.89 | 54.12 | 41.32 | 31.64 | 24.66 |
| TwoStream-SLT-K | 53.42 | 53.68 | 40.83 | 31.22 | 24.43 | 53.95 | 54.24 | 41.34 | 31.64 | 24.58 |
| TwoStream-SLT-J | 54.00 | 54.39 | 41.25 | 31.50 | 24.61 | 54.45 | 54.73 | 41.54 | 31.68 | 24.64 |
| TwoStream-SLT-V* | 54.67 | 54.66 | 41.79 | 32.21 | 25.28 | 55.18 | 54.75 | 41.90 | 32.21 | 25.15 |
| TwoStream-SLT-K* | 54.03 | 54.43 | 41.60 | 31.95 | 25.01 | 55.07 | 55.34 | 42.36 | 32.58 | 25.42 |
| TwoStream-SLT-J* | 54.26 | 54.84 | 41.79 | 32.01 | 24.95 | 54.94 | 54.91 | 42.04 | 32.32 | 25.25 |
| TwoStream-SLT | **55.10** | **55.21** | **42.31** | **32.71** | **25.76** | **55.72** | **55.44** | **42.59** | **32.87** | **25.79** |

the probability for the $i$-th token as

$$\frac{1}{3}\left(p^v(s_i|s_{<i},\mathcal{V}) + p^k(s_i|s_{<i},\mathcal{V}) + p^j(s_i|s_{<i},\mathcal{V})\right), \tag{6}$$

where $p^v$, $p^k$, and $p^j$ denote predictions from the translation networks appended to the video head, keypoint head, and joint head, respectively. The averaged probabilities are used to decode text sequences. We use a beam search decoder with a beam width of 5 and without length penalty.

### B.3 Keypoint Illustration.

We show the keypoints used in our TwoStream network in Figure 1. There are 79 keypoints in total, including 26 face keypoints, 42 hand keypoints, and 11 upper body keypoints.

## C More Experiments

**Ablation Study of Sign Language Translation.** We show how our two-stream network can boost the performance of sign language translation in Table 1. We leverage a network containing a single video encoder followed by an MLP and a translation network as our baseline. We name this network as SingleStream-SLT. Note that our TwoStream-SLT appends an independent translation network onto the video head, keypoint head, and joint head, respectively. For a fair comparison with the baseline, we present three variants of TwoStream-SLT, namely TwoStream-SLT-V, TwoStream-SLT-K, and TwoStream-SLT-J. In each of the variants, only a single translation network is appended onto the video head, keypoint head, or joint head. We observe that almost all variants achieve comparable or better performance than SingleStream-SLT, showing the superiority of the two-stream design. To further demonstrate the benefits of our translation fusion strategy, we separately train

Table 2: Ablation study on the effects of each proposed component in TwoStream-SLR on Phoenix-2014 and CSL-Daily datasets. The evaluation metric is WER in %.

| Dataset | RGB | Keypoints | Bilateral | Pyramid | Joint Head | Distillation | Dev | Test |
|---|---|---|---|---|---|---|---|---|
| Phoenix-2014 | ✓ | | | | | | 22.44 | 23.32 |
| | | ✓ | | | | | 28.56 | 28.00 |
| | ✓ | ✓ | | | | | 22.02 | 22.76 |
| | ✓ | ✓ | ✓ | | | | 19.89 | 20.27 |
| | ✓ | ✓ | ✓ | ✓ | | | 19.42 | 19.62 |
| | ✓ | ✓ | ✓ | ✓ | ✓ | | 18.97 | 19.04 |
| | ✓ | ✓ | ✓ | ✓ | ✓ | ✓ | **18.39** | **18.79** |
| CSL-Daily | ✓ | | | | | | 28.88 | 28.50 |
| | | ✓ | | | | | 34.59 | 34.08 |
| | ✓ | ✓ | | | | | 28.44 | 28.43 |
| | ✓ | ✓ | ✓ | | | | 27.55 | 27.01 |
| | ✓ | ✓ | ✓ | ✓ | | | 26.50 | 26.18 |
| | ✓ | ✓ | ✓ | ✓ | ✓ | | 25.89 | 25.47 |
| | ✓ | ✓ | ✓ | ✓ | ✓ | ✓ | **25.44** | **25.33** |

three TwoStream-SLT-V/K/J models with different random seeds and average predictions of the three translation networks attached to the same head. The model with the fusion strategy is termed as TwoStream-SLT-V*/K*/J*. Our TwoStream-SLT outperforms all three models with the fusion strategy showing that the improvement comes not only from the ensemble strategy, but also from our two-stream visual encoder.

**Ablation Study on Phoenix-2014 and CSL-Daily.** We conduct extra studies to show the effects of each proposed component on Phoenix-2014 and CSL-Daily SLR tasks as shown in Table 2.

## D   Qualitative Results

**Keypoint Heatmaps Generated by HRNet.** As shown in Figure 2, we visualize the keypoint heatmaps generated by HRNet by randomly selecting five frames from the dev set of Phoenix-2014T and CSL-Daily, respectively. We find that the heatmaps are robust to signer appearance, hand positions, and palm orientation in most cases.

**SLR Results.** As shown in Table 3, we conduct qualitative analysis for our TwoStream-SLR and show three samples from the dev set of Phoenix-2014T and CSL-Daily, respectively. It is clear to see that using both video and keypoint streams (TwoStream-SLR) yields more accurate gloss predictions than using either one of them. The result suggests the effectiveness of integrating the features extracted from both the streams.

## E   Broader Impact and Limitations

In this paper, we propose a two-stream network for sign language recognition (SLR) and sign language translation (SLT) to improve communication between the hearing people and the deaf community. Deep learning benefits from large-scale training data while existing datasets of SLR and SLT only contain thousands of parallel data, which may make the training insufficient. There may be unpredictable failures, similar to most translation systems. Please do not use it for scenarios where failures will lead to serious consequences. The method is data-driven, and thus the performance may be affected by biases in the data. So please also be careful about the data collection process when using it. In addition, our approach relies on the keypoints estimator. Inaccurate estimations may hurt the performance. Improving the keypoints estimator is another promising way to facilitate sign language recognition and translation in our framework.

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

Table 3: Qualitative results on Phoenix-2014T (Example (a,b,c)) and CSL-Daily (Example (d,e,f)). We use different colors to represent substitutions, deletions, and insertions, respectively.

| Example (a) | | WER |
|---|---|---|
| Groundtruth | SUED MEHR RUHIG FREUNDLICH BISSCHEN MILD (South More Calm Friendly Little Mild) | - |
| Pred. (TwoStream-SLR) | SUED MEHR RUHIG FREUNDLICH BISSCHEN MILD (South More Calm Friendly Little Mild) | 0.00 |
| Pred. (Video only) | SUED MEHR RUHIG FREUNDLICH BISSCHEN MILD (South More Calm Friendly Little Mild) | 16.67 |
| Pred. (Keypoint only) | SUED MEHR RUHIG FREUNDLICH TATSAECHLICH BESSER (South More Calm Friendly Indeed Better) | 33.33 |

| Example (b) | | WER |
|---|---|---|
| Groundtruth | AUCH NAH FLUSS UND ALPEN AUCH MOEGLICH NEBEL (Also Close Flow And Alps Also Possible Fog) | - |
| Pred. (TwoStream-SLR) | AUCH NAH FLUSS UND BERG ALPEN AUCH MOEGLICH NEBEL (Also Close Flow And Mountain Alps Also Possible Fog) | 12.50 |
| Pred. (Video only) | AUCH NAH FLUSS UND ALPEN BERG AUCH MOEGLICH NEBEL (Also Close Flow And Alps Mountain Also Possible Fog) | 25.00 |
| Pred. (Keypoint only) | AUCH NAH BISSCHEN BERG TAL AUCH MOEGLICH NEBEL (Also Close Little Mountain Valley Also Possible Fog) | 37.50 |

| Example (c) | | WER |
|---|---|---|
| Groundtruth | NACHT SYLT DREIZEHN GRAD MITTE BERG TAL NULL GRAD (Night Sylt Thirteen Degree Center Mountain Valley Zero Degree) | - |
| Pred. (TwoStream-SLR) | NACHT SYLT DREIZEHN GRAD MITTE BERG TAL NULL GRAD (Night Sylt Thirteen Degree Center Mountain Valley Zero Degree) | 0.00 |
| Pred. (Video only) | NACHT SYLT DREIZEHN GRAD MITTE BERG TAL NULL GRAD (Night Sylt Thirteen Degree Center Mountain Valley Zero Degree) | 11.11 |
| Pred. (Keypoint only) | NACHT DONNERSTAG DREIZEHN GRAD MITTE BERG TAL NULL GRAD (Night Thursday Thirteen Degree Center Mountain Valley Zero Degree) | 22.22 |

| Example (d) | | WER |
|---|---|---|
| Groundtruth | 因为 天气 不好 飞机 取消 (Because Weather Bad Flight Canceled) | - |
| Pred. (TwoStream-SLR) | 因为 天气 不好 飞机 取消 (Because Weather Bad Flight Canceled) | 0.00 |
| Pred. (Video only) | 因为 天气 不好 飞机 取消 (Because Weather Bad Flight Canceled) | 20.00 |
| Pred. (Keypoint only) | 因为 天气 不好 看 删除 (Because Weather Bad Look Deleted) | 40.00 |

| Example (e) | | WER |
|---|---|---|
| Groundtruth | 泡 脚 冬天 好 (Bath Feet Winter Good) | - |
| Pred. (TwoStream-SLR) | 泡 脚 冬天 好 (Bath Feet Winter Good) | 0.00 |
| Pred. (Video only) | 泡 脚 冬天 好 (Bath Feet Winter Good) | 25.00 |
| Pred. (Keypoint only) | 泡 脚 冬天 好 (Bath Feet Winter Good) | 25.00 |

| Example (f) | | WER |
|---|---|---|
| Groundtruth | 农民 对 狗 心 爱护 结果 狗 咬 (Farmer To Dog Heart Love Result Dog Bite) | - |
| Pred. (TwoStream-SLR) | 农民 对 狗 心 爱护 最后 结果 狗 咬 (Farmer To Dog Heart Love At last Result Dog Bite) | 12.50 |
| Pred. (Video only) | 农村 对 狗 心 爱护 结果 狗 咬 (Rural To Dog Heart Love Result Dog Bite) | 25.00 |
| Pred. (Keypoint only) | 农村 对 狗 心 爱护 最后 结果 狗 咬 (Rural To Dog Heart Love At last Result Dog Bite) | 37.50 |