# OpenReview forum: "Two-Stream Network for Sign Language Recognition and Translation"
_NeurIPS.cc/2022/Conference — NeurIPS 2022 Accept_

### Official Review · Reviewer_dyUD · 2022-07-08

**Rating:** 9
**Confidence:** 5
**Soundness:** 4 excellent
**Presentation:** 4 excellent
**Contribution:** 4 excellent

**Summary:**

This paper proposes a two-stream network for sign language recognition and translation. The main idea behind this paper is using two separate S3D networks to encode RGB modality and human keypoint modality respectively according to the fact that sign languages use both manual articulations and non-manual elements to convey information. To make the two streams interact with each other, authors propose bidirectional lateral connection, sign pyramid network with auxiliary supervision, and a frame-level self-distillation strategy. Elaborate ablation studies have verified the effectiveness of each proposed component. The results are very solid. TwoStream-SLR which is designed for sign language recognition task, achieves 18.8 WER on Phoenix-2014, 19.0 on Phoenix-2014T and 25.3 on newly published CSL-Daily, which greatly outperforms prior methods by large margins. As for TwoStream-SLT, it also achieves  state-of-the-art performance on Phoenix-2014T and CSL-Daily datasets.

**Questions:**

1. Self-distillation loss brings improvement as shown in Table 3. Can authors study the weight of this distillation loss in Eq 1.
2. As described in L169, authors utilize a Gaussian function to generate keypoint heatmaps. I don’t see studies about sigma, as well as the resolution of heatmaps. I hope authors could provide these studies since how to model keypoints is a key contribution, adding these experiments will make the submission stronger.


**Limitations:**

Due to the data bias and data scarcity issue, there are unpredictable recognition/translation errors as shown in Table 4 in Appendix. Authors also discuss the limitations and societal impact adequately.

**Strengths And Weaknesses:**

Strengths:
1.	This paper is very well written and easy to follow. The motivation is very clear.
2.	The proposed TwoStream network with bidirectional lateral connection, sign pyramid network, auxiliary supervision and frame-level self-distillation is technically sound. I believe this paper will facilitate this research direction.
3.	System-level experiments are very solid. This paper achieves state-of-the-art performance on two sign language understanding tasks, i.e. continuous sign language recognition and sign language translation across several datasets. It is worth mentioning that TwoStream-SLR outperforms previous best methods by large margins, especially on CSL-Daily dataset.
4.	A variety of ablation studies verify the effectiveness of each proposed component, as shown in Table 3 and 4.(a)-(d), as well as the Tables in appendix.

Weakness:
1.	Authors could move the formulation of CTC loss and translation loss from Appendix to the main paper to make the paper clearer.

---

> ### Author Response · Authors · 2022-08-02
> **Responses to Reviewer dyUD**
>
> Thanks for your constructive comments. Our responses to them are given below.
>
> **Q1**
> *Move the formulation of CTC loss and translation loss from Appendix to the main paper.*
>
> Thanks for your suggestion. We will move the formulation into the main paper in our final version.
>
>
> **Q2**
> *Study the weight of the distillation loss in Eq 1.*
>
> We study the weight of the distillation loss $\mathcal{L}\_{dist}$ in Table 3c in the supplementary material. This study is conducted on the Phoenix-2014T SLR task. We redraw the table below. The weight of the self-distillation loss controls the trade-off between the pseudo fine-grained supervision (self-distillation loss) and the coarse-grained supervision (CTC loss). In our experiment, we find that the best performance is obtained when the weight is set to 1.0.
> | Weight of $\mathcal{L}\_{dist}$ | Dev   | Test  |
> |:---------:|:-------:|:-------:|
> | 0.2     | 18.41 | 19.21 |
> | 0.5     | 18.20 | 19.63 |
> | 1.0     | **17.48** | **19.04** |
> | 1.5     | 18.28 | 19.93 |
> | 2.0     | 17.83 | 19.28 |
>
> **Q3**
> *The effect of $\sigma$ and resolution of keypoint heatmaps.*
>
> We study the effect of $\sigma$ and resolution of the keypoints heatmaps in Table 3b in the supplementary material. We redraw the table below.  We find that decreasing heatmap resolution hurts performance while varying $\sigma$ has minor effect. In our experiment, $\sigma=4$ and $H'=W'=112$ can achieve the best performance.
> | $\sigma$ | ($H'$,$W'$) | Dev   | Test |
> |:-------:|:---------:|:-------:|:-------:|
> | 1     | 56      | 29.78 | 28.72 |
> | 2     | 56      | 30.10 | 29.02 |
> | 2     | 112     | 27.22 | 27.52 |
> | 4     | 112     | **27.14** | 27.19 |
> | 6     | 112     | 27.94 | **27.10** |

---

> > ### Comment · Reviewer_dyUD · 2022-08-05
> > **Update my score**
> >
> > I read the response and comments from other reviewers. The rebuttal has addressed all my concerns, as well as many points raised by other reviewers. This is a solid work with convincing experiments and studies. The newly added experiments to verity signer-independentrecognition and background change make the submission stronger. In light of this, I raise my score. I look forward to seeing the additional experiments in the revised paper.

---

### Official Review · Reviewer_Gxi4 · 2022-07-11

**Rating:** 5
**Confidence:** 4
**Soundness:** 2 fair
**Presentation:** 3 good
**Contribution:** 2 fair

**Summary:**

In the SLR task, the paper introduces two separate streams to model both the raw videos and the keypoint sequences. For the two streams to interact with each other better, the paper proposes a variety of techniques, including bidirectional lateral connection, sign pyramid network, and frame-level self-distillation. In the SLT task, the paper extends TwoStream-SLR to TwoStream-SLT by attaching an MLP and NMT. Experimental results show that TwoStream-SLR and TwoStream-SLT achieve SOTA performance on SLR and SLT tasks in three datasets.

**Questions:**

1. The logical coherence of the abstract needs strengthening. For example, in line4-line7, the link between the questions about visual redundancy and the need to incorporate domain knowledge is rather jumpy, thus affecting the fluency of the reading.
2. In the 3.1 section, the details of the selection of the key point are not clear enough. Why were the 79 key points chosen and what were the reasons? The choice of key points was not explored further in the experiment part.
3. In the SLT experiments, in past studies, the results of experiments with added gloss supervision were higher than the gloss-free method. However, the opposite results were obtained in this experiment. It is necessary to analyze and explain the reasons for this.
4. Suggest that some methods of the visual field need to provide an explanation to increase the readability of the paper.

**Ethics Review Area:**

["I don’t know"]

**Strengths And Weaknesses:**

Strengths:

1. The paper proposes Two-Stream Network, including bidirectional lateral connection, sign pyramid network, and frame-level self-distillation methods, to interact RGB videos and keypoint sequences for advancing SLR and SLT.
2. The paper reported an improvement on SLR and SLT tasks.

Weaknesses:

1. These lack the evaluation of visual redundancy and interact with both the raw videos and the keypoint sequences.

2. It is unclear about the usage of domain knowledge and key sequences.

---

> ### Author Response · Authors · 2022-08-02
> **Responses to Reviewer Gxi4 (Part 1)**
>
> Thanks for your constructive comments. Our responses to them are given below.
>
> **Q1**
> *Evaluation of visual redundancy and interactions between the raw videos and keypoint sequences.*
>
> First, we give more explanations about visual redundancy and provide evaluations. Sign languages are visual languages that use two types of features to convey information: manual elements including handshape, palm orientation, movement, and location; non-manual markers such as facial expression and movement of the body, head (nod/shake/tilt), and mouth (mouthing). Raw videos inherently provide rich and complete information for sign language understanding; however, irrelevant factors (e.g. background) which are always regarded as visual redundancy, are also included. To reduce the negative effects raised by visual redundancy, we propose to model keypoints which are more robust to background change. We conduct extra experiments to verify this. Concretely, we use Mask2Former [1] to segment the signer (foreground) appeared in each test video from Phoenix-2014/Phoenix-2014T/CSL-Daily and then paste the estimated foreground onto a few pre-defined backgrounds to synthesize test videos. These new test videos have backgrounds which are unseen in training samples. We experiment on different new backgrounds including uniformly-colored canvas (white, black, green, and blue) and several clutter scenes (studios and street). We compare our single-stream SLR with only RGB inputs with our two-stream network to verify the generalization capability when background changes. The results are shown in the following table. Replacing the backgrounds of original test videos with new ones leads to performance degradation for both methods. However, TwoStream-SLR can effectively reduce the discrepancy, achieving smaller performance gap between new test set and the original test set. For example, when evaluated on Phoenix-2014 test videos with a studio background, the WER of Video-only rises dramatically from 23.32\% to 30.31\% (+6.99\%) while TwoStream-SLR still performs well with WER rising from 18.79\% to 21.85\% (+3.06\%). This experiment proves that our method is less sensitive to background mismatch. We will add more discussion in our revised version.
>
> |**Phoenix-2014** |  **Video only** | | TwoStream-SLR| |
> |------------|:--------------:|:--------------:|:------------|:--------------:|
> | Background | Dev/Test (WER) | Dev/Test (gap) | Dev/Test (WER) | Dev/Test (gap) |
> | original   |   22.44/23.32  |      --/--     | 18.39/18.79 | --/-- |
> | white      |   24.84/25.94  |    2.40/2.62   | 19.91/20.36 | 1.52/1.57 |
> | green      |   24.26/25.63  |    1.82/2.31   | 20.13/19.99 | 1.74/1.20 |
> | blue       |   24.82/25.30  |    2.38/1.98   | 18.90/19.51 | 0.51/0.72 |
> | black      |   23.97/24.44  |    1.53/1.12   | 19.80/19.93 | 1.41/1.14 |
> | studio \#0 |   30.93/30.31  |    8.49/6.99   | 21.46/21.85 | 3.07/3.06 |
> | studio \#1 |   25.06/26.15  |    2.62/2.83   | 20.04/20.07 | 1.65/1.28 |
> | street     |   76.53/74.96  |   54.09/51.64  | 33.91/33.77 | 15.52/14.98 |
>
> |**Phoenix-2014T** |  **Video only** | | TwoStream-SLR| |
> |------------|:--------------:|:--------------:|:------------|:--------------:|
> | Background | Dev/Test (WER) | Dev/Test (gap) | Dev/Test (WER) | Dev/Test (gap) |
> | original   |   21.08/22.42  |      --/--     | 17.48/19.04 | --/-- |
> | white      |   22.47/24.54 | 1.39/2.12 | 18.65/19.96 | 1.17/0.92 |
> | green      |   21.88/24.16 | 0.80/1.74 | 18.44/19.75 | 0.96/0.71 |
> | blue       |   22.68/23.41 | 1.60/0.99 | 18.04/19.65 | 0.56/0.61 |
> | black      |   21.67/23.06 | 0.59/0.64 | 18.12/19.07 | 0.64/0.03 |
> | studio \#0 |   27.89/27.94 | 6.81/5.52 |20.12/20.87 | 2.64/1.83 |
> | studio \#1 |   22.68/23.64 | 1.60/1.22 | 18.52/19.61 | 1.04/0.57 |
> | street     |   74.83/73.02 | 53.75/50.60 | 37.55/36.65 | 20.07/17.61 |
>
> |**CSL-Daily** |  **Video only** | | TwoStream-SLR| |
> |------------|:--------------:|:--------------:|:------------|:--------------:|
> | Background | Dev/Test (WER) | Dev/Test (gap) | Dev/Test (WER) | Dev/Test (gap) |
> | original   | 28.88/28.50 |  --/--  |  25.44/25.33 | --/-- |
> | white      |   29.76/29.25 | 0.88/0.75 | 25.91/25.91 | 0.47/0.58 |
> | green      |   29.65/29.40 | 0.77/0.90 | 26.23/25.94 | 0.79/0.61 |
> | blue       |  31.59/31.23 | 2.71/2.73 | 25.95/26.04 | 0.51/0.71 |
> | black      |  29.30/29.24 | 0.42/0.74 | 25.79/25.71 | 0.35/0.38 |
> | studio \#0 |  47.31/47.17 | 18.43/18.67 | 28.91/28.99 | 3.47/3.66 |
> | studio \#1 |  30.65/30.35 | 1.77/1.85 |  26.23/26.43 | 0.79/1.10 |
> | street     |  91.97/92.49 | 63.09/63.99 | 62.71/62.00 | 37.27/36.67 |
>
> **References**
>
> [1] Bowen Cheng, Ishan Misra, Alexander G. Schwing, Alexander Kirillov, and Rohit Girdhar. Masked attention mask transformer for universal image segmentation. In CVPR, 2022.

---

> ### Author Response · Authors · 2022-08-02
> **Responses to Reviewer Gxi4 (Part 2)**
>
> **Q1**
> *Evaluation of visual redundancy and interactions between the raw videos and keypoint sequences.*
>
> Next, we respond to the evaluation of interactions between two streams. As we stated in L59-68 in the main paper, motion blur in sign videos and domain gap between the COCO-WholeBody training set and sign language recognition datasets lead to inaccurate keypoints, which motivates us to jointly model RGB videos and keypoint sequences to complement each other. To this end, we propose bidirectional lateral connection, joint head, and frame-level self-distillation to make the two streams interactive and promote each other. The effects of each proposed component are verified in Table 3 in the main paper. Meanwhile, we conduct exhaustive analysis in L273-283. Here we redraw the table below:
>
> | V-Encoder  | K-Encoder  | Bilateral | SPN | Joint-Head | Distillation | Dev | Test |
> |:------------:|:------------:|:-----------:|:-----:|:------------:|:--------------:|:-----:|:------|
> |$\checkmark$|                       |                        |                      |                        |                      |  21.08  | 22.42 |
> |                       | $\checkmark$|                       |                      |                        |                      |  27.14  | 27.19 |
> |$\checkmark$|$\checkmark$|                       |                      |                        |                      |  20.47  | 21.55 |
> |$\checkmark$|$\checkmark$|$\checkmark$|                       |                        |                      |  19.03  | 20.12 |
> |$\checkmark$|$\checkmark$|$\checkmark$|$\checkmark$|                       |                      |  18.52   | 19.91 |
> |$\checkmark$|$\checkmark$|$\checkmark$|$\checkmark$|$\checkmark$|                      |  18.36   | 19.49 |
> |$\checkmark$|$\checkmark$|$\checkmark$|$\checkmark$|$\checkmark$|$\checkmark$|  **17.48**   | **19.04** |
>
> Particularly, we conduct ablation studies on the lateral connections as shown in Table 4(a) in our main paper. Specifically, we train models with only unilateral connections (video $\rightarrow$ keypoint or keypoint $\rightarrow$ video) or reduce the number of features on which the bilateral connection is applied. We compare these variants to our final model. We redraw the table as below. It can be seen that performing bilateral connection on C1, C2, and C3 achieves the best results, verifying that more two-stream interactions lead to better performance.
>
> |V$\rightarrow$K|K$\rightarrow$V|Connection|Dev|Test|
> |:---------------------:|:---------------------:|:----------------:|:-----:|:-------:|
> |                        |                        |         None |   18.57 | 20.03|
> |       $\checkmark$                 |                        |    C1,C2,C3 |   17.88 | 19.61 |
> |               |         $\checkmark$               |    C1,C2,C3 |   18.82 | 19.93 |
> |        $\checkmark$           |         $\checkmark$               |    C1,C2,C3 |   **17.48** | **19.04** |
> |        $\checkmark$           |         $\checkmark$               |    C2,C3 |   17.59| 19.30 |
>
> Thanks for your comment and we will add more discussions in our revised version.

---

> ### Author Response · Authors · 2022-08-02
> **Responses to Reviewer Gxi4 (Part 3)**
>
> **Q2**
> *It is unclear about the usage of domain knowledge and keypoint sequences.*
>
> We state the domain knowledge of sign languages in L36-39 "Sign languages use two types of visual signals to convey information: manual elements that include handshape, palm orientation, etc., and non-manual markers such as facial expressions and movement of the body, head, mouth, eyes, and eyebrows". How to leverage such domain knowledge is still underexplored. In our work, we propose to model human keypoint sequences, which contain key information for sign language understanding, to inject "inductive bias" and "domain knowledge" to ease the learning. Please also refer to our responses to your first question for the benefits of modeling keypoints.
>
> **Q3**
> *The logical coherence of the abstract needs strengthening.*
>
> Thanks for your suggestion. We will rephrase the statement in our final version.
>
> **Q4**
> *The details of the selection of the keypoints are not clear enough in Section 3.1.*
>
> HRNet trained on COCO-WholeBody could generate 11 upper body keypoints, 42 hand keypoints, 20 mouth keypoints, and 48 face keypoints (121 keypoints in total). In our main paper, we use all upper body and hand keypoints, but reduce the other keypoints by spatially evenly sampling 10 mouth keypoints and 16 face keypoints (79 keypoints in total) as a trade-off between accuracy and computational costs.
>
> Due to the limited space of the main paper, we include the ablation study of the choice of keypoints in Table 3a in our supplementary material. We redraw the table below. We train single-stream SLR models with different keypoints choices as inputs. All studies are conducted on Phoenix-2014T SLR task. We observe that all parts contribute to sign language recognition.
> | Upper-body   | Hand         | Mouth        | Face         | #Keypoints | Dev   | test  |
> |:--------------:|:--------------:|:--------------:|:--------------:|:------------:|:-------:|:-------:|
> | $\checkmark$ |              |              |              | 11         | 49.11 | 48.46 |
> | $\checkmark$ | $\checkmark$ |              |              | 53(+42)    | 37.15 | 36.88 |
> | $\checkmark$ | $\checkmark$ | $\checkmark$ |              | 63(+10)    | 28.42 | 28.20 |
> | $\checkmark$ | $\checkmark$ | $\checkmark$ | $\checkmark$ | 79(+16)    | **27.14** | **27.19** |
>
> To further explore the choice of keypoints, we conduct more experiments by varying the number of keypoints of each part. We show the results as follows. It can be seen that our model is insensitive to the choice of keypoints. We will add this experiment in our revised version.
>
> | #Upper-body(11) | #Hand(42) | #Mouth(20) | #Face(48) | #Total(121) | Dev   | Test  |
> |:-----------------:|:-----------:|:------------:|:-----------:|:-------------:|:-------:|:-------:|
> | 11              | 42        | 10         | 16        | 79         | **27.14** | 27.19 |
> | 11              | 42        | 20(+10)    | 16        | 89 (+10)    | 27.36 | 27.07 |
> | 11              | 42        | 10         | 24(+8)    | 87(+8)      | 27.25 | 27.00 |
> | 11              | 42        | 10         | 48(+32)   | 111(+32)    | 27.25 | 27.24 |
> | 11              | 42        | 20(+10)    | 48(+32)   | 121(+42)    | 27.22 | 27.21 |
> | 6(-5)           | 42        | 10         | 16        | 74(-5)      | 27.25 | 27.85 |
> | 11              | 21(-21)   | 10         | 16        | 58(-21)     | 27.33 | 27.19 |
> | 11              | 42        | 5(-5)      | 16        | 74(-5)      | 27.78 | 28.06 |
> | 11              | 42        | 10         | 8(-8)     | 71(-8)      | 27.65 | 27.57 |

---

> ### Author Response · Authors · 2022-08-02
> **Responses to Reviewer Gxi4 (Part 4)**
>
> **Q5**
> *In the SLT experiments, in past studies, the results of experiments with added gloss supervision were higher than the gloss-free method. However, the opposite results were obtained in this experiment.*
>
> We firstly clarify that gloss-supervised methods do outperform gloss-free methods in our paper. We conjecture that your ``gloss-free'' refers to the Sign2Text in Table 2 of the main paper. Sign2Text is a task of directly translating sign videos into natural languages with or without intermediate gloss supervision. Our method, as well as MMTLB [2], SignBT [3], STMC-T [4], and Joint-SLRT [5] all utilize the gloss supervision to facilitate sign language translation, and surpass gloss-free Sign2Text methods including TSPNet [6] and SL-Luong [7] by large margins (see Sign2Text part in Table 2 of the main paper). For example, our approach (with gloss supervision) outperforms TSPNet (without gloss supervision) by 15.54 BLEU-4 on Phoenix-2014T test set. We also mark gloss-free methods (TSPNet and SL-Luong) in Table 2 of the main paper. Please correct us if our conjecture is wrong.
>
> Next, we make clarification that our Sign2Text (with gloss supervision) model outperforms the Sign2Gloss2Text one. Previous works [3, 5, 7] categorize sign language translation into two types, namely Sign2Text and Sign2Gloss2Text. Sign2Text directly generates spoken texts from given sign videos, where gloss annotations are optional to supervise intermediate representations and better performance is observed if gloss supervision is introduced. Sign2Gloss2Text, however, is a two-stage framework that first uses a sign language recognition system to predict glosses from sign videos (thus gloss annotations are required) and then translates gloss sequence into spoken language. Our work as well as some previous works [2, 3, 5] show that Sign2Text outperforms Sign2Gloss2Text. We suspect this is because Sign2Gloss2Text uses discrete glosses as intermediate representations while Sign2Text processes dense visual features to better capture rich spatial-temporal semantics in sign videos. Besides, Sign2Text mitigates error propagation which hinders performance of Sign2Gloss2Text.
>
> Thanks for your comments on this. We will improve the description of Sign2Text in our revised version.
>
>
> **Q6**
> *Explanations for the methods of the visual field.*
>
> Thanks for your suggestion. We will refine our paper to increase the readability of our work.
>
> **References**
>
> [1] Bowen Cheng, Ishan Misra, Alexander G. Schwing, Alexander Kirillov, and Rohit Girdhar. Masked attention mask transformer for universal image segmentation. In CVPR, 2022.
>
> [2] Yutong Chen, Fangyun Wei, Xiao Sun, Zhirong Wu, and Stephen Lin. A simple multi-modality transfer learning baseline for sign language translation. arXiv preprint arXiv:2203.04287, 2022.
>
> [3] Hao Zhou, Wengang Zhou, Weizhen Qi, Junfu Pu, and Houqiang Li. Improving sign language translation with monolingual data by sign back-translation. In Proceedings of the IEEE/CVF Conference on Computer Vision and Pattern Recognition, 2021.
>
> [4] Hao Zhou, Wengang Zhou, Yun Zhou, and Houqiang Li. Spatial-temporal multi-cue network for sign language recognition and translation. IEEE Transactions on Multimedia, 2021.
>
> [5] Necati Cihan Camgoz, Oscar Koller, Simon Hadfield, and Richard Bowden. Sign language transformers: Joint end-to-end sign language recognition and translation. In Proceedings of the IEEE/CVF conference on computer vision and pattern recognition, 2020.
>
> [6] Dongxu Li, Chenchen Xu, Xin Yu, Kaihao Zhang, Benjamin Swift, Hanna Suominen, and Hongdong Li. Tspnet: Hierarchical feature learning via temporal semantic pyramid for sign language translation. In Advances in Neural Information Processing Systems, 2020.
>
> [7]  Necati Cihan Camgoz, Simon Hadfield, Oscar Koller, Hermann Ney, and Richard Bowden. Neural sign language translation. In Proceedings of the IEEE Conference on Computer Vision and Pattern Recognition, 2018.

---

### Official Review · Reviewer_rULN · 2022-07-11

**Rating:** 7
**Confidence:** 3
**Soundness:** 3 good
**Presentation:** 4 excellent
**Contribution:** 3 good

**Summary:**

This paper proposes a two-stream model including the raw videos and the keypoint sequences for sign language recognition and sign language translation. The two-stream model also utilize a variety of techniques, including bidirectional lateral connection, sign pyramid network, and frame-level self-distillation to further improve their model. Empirically, the two-stream model outperforms the previous best method.

**Questions:**

1. It may be confused that why the keypoint sequences can achieve such an effect improvement.
2. This paper introduce the robustness of previous models, which suffer from dramatic performance degradation when the background or signer is mismatched between training and testing, but is the two-stream model can solve this problem?

**Limitations:**

The authors have addressed the limitations of their work.

**Strengths And Weaknesses:**

Strengths:
1. This paper takes advantage of the keypoint sequences and a variety of techniques and achieve excellent results.
2. This paper is well written.
Weaknesses:
1. It may be confused that why the keypoint sequences can achieve such an effect improvement.
2. This paper introduce the robustness of previous models, which suffer from dramatic performance degradation when the background or signer is mismatched between training and testing, but is the two-stream model can solve this problem?

---

> ### Author Response · Authors · 2022-08-02
> **Responses to Reviewer rULN (Part 1)**
>
> Thanks for your constructive comments. Our responses to them are given below.
>
> **Q1**
> *Why can the keypoint sequences achieve such an improvement?*
>
> There are two reasons:
>
> First, sign languages are visual languages that use two types of features to convey information: manual elements including handshape, palm orientation, movement, and location; non-manual markers such as facial expression and movement of the body, head (nod/shake/tilt), and mouth (mouthing). All these features need to be considered to capture the complete meanings of sign languages. The keypoint stream in our TwoStream Network can fully incorporate the above domain knowledge to promote sign language recognition and translation. Concretely, we take into account manual elements by modeling 42 hand keypoints, and non-manual markers by modeling 11 upper body keypoints, 10 mouth keypoints, and 16 face keypoints. The illustration of keypoints used in our approach is shown in Figure 1 in our supplementary material.
>
> Second, sign language recognition and translation suffer from the data scarcity issue. Training an efficient neural machine translation model often requires a corpus of 1M parallel data, however, existing sign language recognition datasets only contain 7-20K training samples. Though raw videos provide rich and complete visual information, training on insufficient data may lead the network to overlook the key information mentioned in the first point. Introducing keypoints into learning injects "inductive bias'' and "prior knowledge'' to sign language understanding.
>
> Experiments in Table 3 of the main paper verify the effectiveness of modeling keypoints and interactions between two streams. We redraw the table below.  (The experiment is conducted on Phoenix-2014T SLR task. V: video, K: keypoint, Bilateral: bidirectional lateral connection, SPN: sign pyramid network.)  It shows that simply averaging the predictions of the video and keypoint streams can achieve better performance than only using either one single stream. Besides, the proposed techniques (bidirectional lateral connection, joint head, and cross-distillation) can strengthen the interactions between the two streams and further improve the performance.
> | V-Encoder  | K-Encoder  | Bilateral | SPN | Joint-Head | Distillation | Dev | Test |
> |:------------:|:------------:|:-----------:|:-----:|:------------:|:--------------:|:-----:|:------:|
> |$\checkmark$|                       |                        |                      |                        |                      |  21.08  | 22.42 |
> |                       | $\checkmark$|                       |                      |                        |                      |  27.14  | 27.19 |
> |$\checkmark$|$\checkmark$|                       |                      |                        |                      |  20.47  | 21.55 |
> |$\checkmark$|$\checkmark$|$\checkmark$|                       |                        |                      |  19.03  | 20.12 |
> |$\checkmark$|$\checkmark$|$\checkmark$|$\checkmark$|                       |                      |  18.52   | 19.91 |
> |$\checkmark$|$\checkmark$|$\checkmark$|$\checkmark$|$\checkmark$|                      |  18.36   | 19.49 |
> |$\checkmark$|$\checkmark$|$\checkmark$|$\checkmark$|$\checkmark$|$\checkmark$|  **17.48**   | **19.04** |
>
> Particularly, we conduct ablation studies on the lateral connections as shown in Table 4(a) in our main paper. Specifically, we train models with only unilateral connections (video $\rightarrow$ keypoint or keypoint $\rightarrow$ video) or reduce the number of features on which the bilateral connection is applied. We compare these variants to our final model. We redraw the table as below. It can be seen that performing bilateral connection on C1, C2, and C3 achieves the best results, verifying that more two-stream interactions lead to better performance.
>
> |V$\rightarrow$K|K$\rightarrow$V|Connection|Dev|Test|
> |:---------------------:|:---------------------:|:----------------:|:-----:|:-------:|
> |                        |                        |         None |   18.57 | 20.03|
> |       $\checkmark$                 |                        |    C1,C2,C3 |   17.88 | 19.61 |
> |               |         $\checkmark$               |    C1,C2,C3 |   18.82 | 19.93 |
> |        $\checkmark$           |         $\checkmark$               |    C1,C2,C3 |   **17.48** | **19.04** |
> |        $\checkmark$           |         $\checkmark$               |    C2,C3 |   17.59| 19.30 |
>
> Thanks for your comment and we will add more discussions in our revised version.

---

> ### Author Response · Authors · 2022-08-02
> **Responses to Reviewer rULN (Part 2)**
>
> **Q2**
> *Can the Two-stream model solve the background and signer mismatch problems?*
>
> Keypoints are represented by heatmaps, which are more robust to video background and signer appearance.
> Visualization of keypoint heatmaps is shown in Figure 2 in the supplementary material. Here we conduct the following experiments to verify the generalization of our method under the setting of signer mismatch and background change.
>
> **Signer mismatch** There are few works exploring signer-independent sign language recognition and translation. Phoenix-2014 provides a signer-independent split named Phoenix-2014-SI, where the Signer05 is excluded from the training set and the dev/test set only contains videos signed by Signer05. Our model is retrained on the Phoenix-2014-SI training set and evaluated on the official dev/test set (unseen signer) and the rest of the Phoenix-2014 dev/test set which is comprised of signers appeared in the training set (seen signers). As shown in the following table, our TwoStream-SLR can achieve a WER of 29.53\% and 29.72\% on the dev and test set of the unseen signer, respectively, and outperforms the previous best method CMA [1] by a large margin. More importantly, compared to the baseline that only uses RGB videos, the performance gap between the seen and unseen signers can drop dramatically with the help of keypoint sequences, which implies that our TwoStream Network can relieve the signer mismatch issue.
>
> **Phoenix-2014-SI:**
> | Method               | Dev/Test (seen) | Dev/Test (unseen) | Dev/Test (gap) |
> |----------------------|:---------------:|:-----------------:|:--------------:|
> | ReSign [2]           |      --/--      |     45.1/44.1     |      --/--     |
> | DNF [3]              |      --/--      |     36.0/35.7     |      --/--     |
> | CMA [1]              |      --/--      |     34.8/34.3     |      --/--     |
> | Video only (ours)    |   24.14/25.03   |    37.42/37.14    |   13.28/12.11  |
> | TwoStream-SLR (ours) | **19.75/20.81** |  **29.53/29.72**  |  **9.78/8.91** |
>
> Furthermore, we imitate the creation of the Phoenix-2014-SI dataset to construct another dataset named Phoenix-2014T-SI, where the Signer05 is excluded from the original Phoenix-2014T training set and the dev and test set only contain videos signed by Signer05. We obtain identical conclusions as shown in the following table.
>
> **Phoenix-2014T-SI:**
> | Method               | Dev/Test (seen) | Dev/Test (unseen) | Dev/Test (gap) |
> |----------------------|:---------------:|:-----------------:|:--------------:|
> | Video only (ours)    |   22.54/24.77   |    35.15/34.80    |   12.61/10.03  |
> | TwoStream-SLR (ours) | **18.25/20.56** |  **27.54/29.03**  |  **9.29/8.47** |
>
> **References**
>
> [1] Junfu Pu, Wengang Zhou, Hezhen Hu, and Houqiang Li. Boosting Continuous Sign Language Recognition via Cross Modality Augmentation, pp. 1497–1505. Association for Computing Machinery, New York, NY,
> 182 USA, 2020.
>
> [2] Oscar Koller, Sepehr Zargaran, and Hermann Ney. Re-sign: Re-aligned end-to-end sequence modelling with deep recurrent cnn-hmms. In IEEE Conference on Computer Vision and Pattern Recognition, 2017.
>
> [3] Runpeng Cui, Hu Liu, and Changshui Zhang. A deep neural framework for continuous sign language recognition by iterative training. IEEE Transactions on Multimedia, 21(7):1880–1891, 2019. doi: 10.1109/174 TMM.2018.2889563.
>
> [4] Bowen Cheng, Ishan Misra, Alexander G. Schwing, Alexander Kirillov, and Rohit Girdhar. Masked attention mask transformer for universal image segmentation. In CVPR, 2022.

---

> ### Author Response · Authors · 2022-08-02
> **Responses to Reviewer rULN (Part 3)**
>
> **Q2**
> *Can the Two-stream model solve the background and signer mismatch problems?*
>
> **Background mismatch**  To verify the robustness of background change, we use Mask2Former [4] to segment the signer (foreground) appeared in each test video from Phoenix-2014/Phoenix-2014T/CSL-Daily and then paste the estimated foreground onto a few pre-defined backgrounds to synthesize test videos. These new test videos have backgrounds which are unseen in training samples. We experiment with different new backgrounds including uniformly-colored canvas (white, black, green, and blue) and several clutter scenes (studio and street). We compare our single-stream SLR with only RGB inputs with our two-stream network to verify the generalization capability when background changes. The results are shown in the following table. Replacing the backgrounds of the original test videos with new ones leads to performance degradation for both methods. However, TwoStream-SLR can effectively reduce the discrepancy, achieving smaller performance gap between the new test sets and the original test set. For example, when evaluated on Phoenix-2014 test videos with a studio background, the WER of Video-only rises dramatically from 23.32\% to 30.31\% (+6.99\%) while TwoStream-SLR still performs reasonably well with WER rising from 18.79\% to 21.85\% (+3.06\%). This experiment proves that our method is less sensitive to background mismatch.
>
>
> |**Phoenix-2014** |  **Video only** | | TwoStream-SLR| |
> |------------|:--------------:|:--------------:|:------------|:--------------:|
> | Background | Dev/Test (WER) | Dev/Test (gap) | Dev/Test (WER) | Dev/Test (gap) |
> | original   |   22.44/23.32  |      --/--     | 18.39/18.79 | --/-- |
> | white      |   24.84/25.94  |    2.40/2.62   | 19.91/20.36 | 1.52/1.57 |
> | green      |   24.26/25.63  |    1.82/2.31   | 20.13/19.99 | 1.74/1.20 |
> | blue       |   24.82/25.30  |    2.38/1.98   | 18.90/19.51 | 0.51/0.72 |
> | black      |   23.97/24.44  |    1.53/1.12   | 19.80/19.93 | 1.41/1.14 |
> | studio \#0 |   30.93/30.31  |    8.49/6.99   | 21.46/21.85 | 3.07/3.06 |
> | studio \#1 |   25.06/26.15  |    2.62/2.83   | 20.04/20.07 | 1.65/1.28 |
> | street     |   76.53/74.96  |   54.09/51.64  | 33.91/33.77 | 15.52/14.98 |
>
> |**Phoenix-2014T** |  **Video only** | | TwoStream-SLR| |
> |------------|:--------------:|:--------------:|:------------|:--------------:|
> | Background | Dev/Test (WER) | Dev/Test (gap) | Dev/Test (WER) | Dev/Test (gap) |
> | original   |   21.08/22.42  |      --/--     | 17.48/19.04 | --/-- |
> | white      |   22.47/24.54 | 1.39/2.12 | 18.65/19.96 | 1.17/0.92 |
> | green      |   21.88/24.16 | 0.80/1.74 | 18.44/19.75 | 0.96/0.71 |
> | blue       |   22.68/23.41 | 1.60/0.99 | 18.04/19.65 | 0.56/0.61 |
> | black      |   21.67/23.06 | 0.59/0.64 | 18.12/19.07 | 0.64/0.03 |
> | studio \#0 |   27.89/27.94 | 6.81/5.52 |20.12/20.87 | 2.64/1.83 |
> | studio \#1 |   22.68/23.64 | 1.60/1.22 | 18.52/19.61 | 1.04/0.57 |
> | street     |   74.83/73.02 | 53.75/50.60 | 37.55/36.65 | 20.07/17.61 |
>
> |**CSL-Daily** |  **Video only** | | TwoStream-SLR| |
> |------------|:--------------:|:--------------:|:------------|:--------------:|
> | Background | Dev/Test (WER) | Dev/Test (gap) | Dev/Test (WER) | Dev/Test (gap) |
> | original   | 28.88/28.50 |  --/--  |  25.44/25.33 | --/-- |
> | white      |   29.76/29.25 | 0.88/0.75 | 25.91/25.91 | 0.47/0.58 |
> | green      |   29.65/29.40 | 0.77/0.90 | 26.23/25.94 | 0.79/0.61 |
> | blue       |  31.59/31.23 | 2.71/2.73 | 25.95/26.04 | 0.51/0.71 |
> | black      |  29.30/29.24 | 0.42/0.74 | 25.79/25.71 | 0.35/0.38 |
> | studio \#0 |  47.31/47.17 | 18.43/18.67 | 28.91/28.99 | 3.47/3.66 |
> | studio \#1 |  30.65/30.35 | 1.77/1.85 |  26.23/26.43 | 0.79/1.10 |
> | street     |  91.97/92.49 | 63.09/63.99 | 62.71/62.00 | 37.27/36.67 |
>
> We will add these experiments and discussions into our revised version.
>
> **References**
>
> [1] Junfu Pu, Wengang Zhou, Hezhen Hu, and Houqiang Li. Boosting Continuous Sign Language Recognition via Cross Modality Augmentation, pp. 1497–1505. Association for Computing Machinery, New York, NY,
> 182 USA, 2020.
>
> [2] Oscar Koller, Sepehr Zargaran, and Hermann Ney. Re-sign: Re-aligned end-to-end sequence modelling with deep recurrent cnn-hmms. In IEEE Conference on Computer Vision and Pattern Recognition, 2017.
>
> [3] Runpeng Cui, Hu Liu, and Changshui Zhang. A deep neural framework for continuous sign language recognition by iterative training. IEEE Transactions on Multimedia, 21(7):1880–1891, 2019. doi: 10.1109/174 TMM.2018.2889563.
>
> [4] Bowen Cheng, Ishan Misra, Alexander G. Schwing, Alexander Kirillov, and Rohit Girdhar. Masked attention mask transformer for universal image segmentation. In CVPR, 2022.

---

### Meta-Review · Area_Chair_gGyb · 2022-09-07

**Recommendation:** Accept
**Confidence:** Less certain

**Metareview:**

This paper extends models for sign language recognition and translation with a dual encoder where, first, keypoint sequences are estimated using an off-the-shelf model, then fused with the video sequence. It is a minor technical contribution to add the keypoint estimations as input since no new information was introduced; however, the authors demonstrated strong execution of experimental results. This paper can be categorized with pipeline/cascade approaches which rely on domain knowledge for engineered feature extraction and combination.
The paper presents many experimental results for architecture changes to improve results: bidirectional lateral connection, sign pyramid network, and frame-level self-distillation.
The authors convinced the reviewers with more experimental results during the rebuttal period leading to two solid, and one borderline accept votes.


**Award:**

No

---

### Decision · Program_Chairs · 2022-09-14

Accept